# Observationally constrained projection of Afro-Asian monsoon precipitation

Ziming Chen[1,2], Tianjun Zhou [1,2,3 ✉], Xiaolong Chen [1,3], Wenxia Zhang [1], Lixia Zhang[1,3], Mingna Wu [1,2] & Liwei Zou[1]

The Afro-Asian summer monsoon (AfroASM) sustains billions of people living in many developing countries covering West Africa and Asia, vulnerable to climate change. Future increase in AfroASM precipitation has been projected by current state-of-the-art climate models, but large inter-model spread exists. Here we show that the projection spread is related to present-day interhemispheric thermal contrast (ITC). Based on 30 models from the Coupled Model Intercomparison Project Phase 6, we find models with a larger ITC trend during 1981–2014 tend to project a greater precipitation increase. Since most models overestimate present-day ITC trends, emergent constraint indicates precipitation increase in constrained projection is reduced to 70% of the raw projection, with the largest reduction in West Africa (49%). The land area experiencing significant increases of precipitation (runoff) is 57% (66%) of the raw projection. Smaller increases of precipitation will likely reduce flooding risk, while posing a challenge to future water resources management.

[1] State Key Laboratory of Numerical Modeling for Atmospheric Sciences and Geophysical Fluid Dynamics, Institute of Atmospheric Physics, Chinese Academy of Sciences, 100029 Beijing, China. [2] University of Chinese Academy of Sciences, 100049 Beijing, China. [3] CAS Center for Excellence in Tibetan Plateau Earth Sciences, Chinese Academy of Sciences (CAS), Beijing, China. ✉email: zhoutj@lasg.iap.ac.cn

The Afro-Asian monsoon, a major component of the global monsoon system, consists of West African, South Asian and East Asian monsoon regions[1–5]. Projection of Afro-Asian summer monsoon (AfroASM) precipitation, which would greatly affect local freshwater resources for billions of people[3,6,7], thus is crucial to climate change adaptation and mitigation activities. Unfortunately, the projections of climate models show a large spread[8–12], hampering the assessment of regional climate change. To yield a more reliable future projection, understanding and reducing the uncertainty are of urgent need.

Great efforts have been devoted to the understanding of the source of uncertainties in monsoon precipitation changes. Multi-model ensemble (MME) projections of CMIP (Coupled Model Intercomparison Project) models suggest that AfroASM precipitation would increase by approximately 8% and 14% under medium and high emission scenarios in the long-term projection, with large inter-model spread of 1%~14% and 3%~25%, respectively[4,11,13–16]. The thermodynamic process related to increases in atmospheric moisture enhances precipitation robustly across models, while the dynamic process associated with changes of circulation contributes to large inter-model uncertainty[7,11]. Total uncertainty in future projection comes from distinct socioeconomic scenarios, stochastic internal variability and different model structures[8,17–21]. The model uncertainty explains more than 70% of total uncertainties of AfroASM precipitation changes in CMIP5 models in the long-term projection[8,22]. The uncertainty can even offset the reliability of monsoon precipitation projection, and hamper the use of this information for policy making[8,23].

In recent years, the emergent constraint technique, which is based on the physical link between a modeled but observable variable in the present day and a projected variable in the future climate system, has been developed to reduce the projection uncertainty and improve the reliability of future projection[24–28]. So far, many factors, such as the SST over cold-tongue regions and the convection over western Pacific, have been used to constrain the CMIP5 projection of summer precipitation over the East Asian, South Asian and West African monsoon regions[9,10,29–31].

In the ongoing CMIP6, while global land monsoon precipitation is projected to increase in the long term under high emission scenarios, there exists large uncertainty at regional scales, in particular over the AfroASM regions[11]. Recent studies reported the connection between the uncertainty of AfroASM precipitation changes and the increase of interhemispheric thermal contrast in the projection[4,12,13,32]. But how to constrain the projection and reduce the spread remains unknown. Given the fact that the monsoon precipitation in West Africa and Asia shows in-phase changes due to the modulation of interhemispheric thermal contrast (ITC) and sea surface temperature (SST) variation of North Atlantic on millennial[1,33], centurial[4,12,32], and decadal timescales[3,34], we hypothesize that the spread of AfroASM precipitation can potentially be constrained by the large scale interhemispheric or land-sea thermal contrast. We examine this hypothesis by using the output of the new Scenario Model Intercomparison Project in CMIP6[35].

In this work, by constraining the spread and biases of ITC in a hierarchical statistical framework, precipitation increase in the constrained projection ($0.57 \pm 0.38$ mm day$^{-1}$, constrained projection with $\pm 1\sigma$ across models) is about 70% of the raw projection, with the largest reduction in the West African monsoon region. About 10% of the inter-model uncertainty in future precipitation changes is reduced. Given that the emergent constraint improves the reliability in AfroASM precipitation projections, we further investigate the impacts of the constrained projection on the potential water availability. The fractions of land area that will experience a significant increase of precipitation and potential water availability are about 57% and 66% of the raw projection, respectively.

## Results

**Dominant uncertainty of AfroASM precipitation projection.** CMIP6 models project a general increase in AfroASM precipitation under the high-emission scenario, Shared Socio-economic Pathways (SSP) 5–8.5 (2050~2099), except for part of the West African monsoon region (Fig. 1; see Methods and Supplementary Table S1). The SSP5-8.5 scenario is a fossil-fuel development pathway, in which the anthropogenic radiative forcing will increase by 8.5 W m$^{-2}$ at the end of 21$^{st}$ century[36–38]. The regional average of the AfroASM precipitation increase is 14% relative to the baseline (1965~2014), with a large inter-model spread (1%~27% for the 5$^{th}$–95$^{th}$ ensemble range; Fig. 1a). The signal-to-noise ratio (SNR), defined as the ratio between the ensemble mean and inter-model standard deviation (STD) of the projected changes, is less than 1.5 over 90% of the Afro-Asian monsoon regions, demonstrating large inter-model uncertainties in the projected AfroASM precipitation (Fig. 1c).

To reveal the sources of model spread, we conduct an inter-model empirical orthogonal function (EOF) analysis on the projected changes of AfroASM precipitation (Fig. 2; see Methods). The leading principal component (PC1) accounts for 26% of inter-model variance (Fig. 2a and Supplementary Fig. S1). The leading uncertainty mode exhibits a spatially consistent increase of precipitation over the AfroASM domain, with a systematic enhancement of monsoon circulation from West Africa, through Indian Peninsular to East Asia (Fig. 2a and Supplementary Fig. S2). To exclude that the above pattern may be dominated by strong diversity in mean precipitation and spatial variability across model, we further scale each model prior to taking the inter-model EOF analysis (see Methods), and obtain similar patterns compared with that in Fig. 2a (not shown). Hence, the synchronized precipitation changes across the entire AfroASM regions in the model spread imply that a large-scale controlling factor may play a dominant role.

**Physical linkage between present-day bias and future projection uncertainty.** The large-scale monsoon circulation is apparently driven by the thermal contrast between Northern Hemisphere (NH) and Southern Hemisphere (SH) due to moist static energy gradients associated with the seasonal swing of solar incidence[39–42]. Given the driving mechanism of monsoon, to understand the leading mode of projection uncertainty, we focus on the warming contrast between NH and SH that drives large-scale monsoon circulation. We regress surface temperature warming in 2050–2099 across models onto the normalized PC1 (Fig. 2b). A robust "NH warmer than SH" pattern is obtained, suggesting that models with larger increases of NH-SH ITC tend to project a wetter AfroASM, which is consistent with the basic driving mechanism of monsoon. In addition, the inter-model spread of NH-SH ITC in future projections correlates with the trend over the historical period (Fig. 2c). This pattern indicates that a model with a larger ITC trend in the present-day climate will project a greater increase of AfroASM precipitation in the future, as shown in the corresponding EOF1 (Fig. 2a). In addition, a remarkable warming anomaly related to PC1 in the historical period is seen over the Southern Ocean, which may be associated with the model biases in sea ice coverage[43].

Why does the significant inter-model correlation between the present-day ITC trend and future AfroASM precipitation exist? A greater NH warming than SH is inherent to global warming given a smaller heat capacity due to a larger land area fraction in the

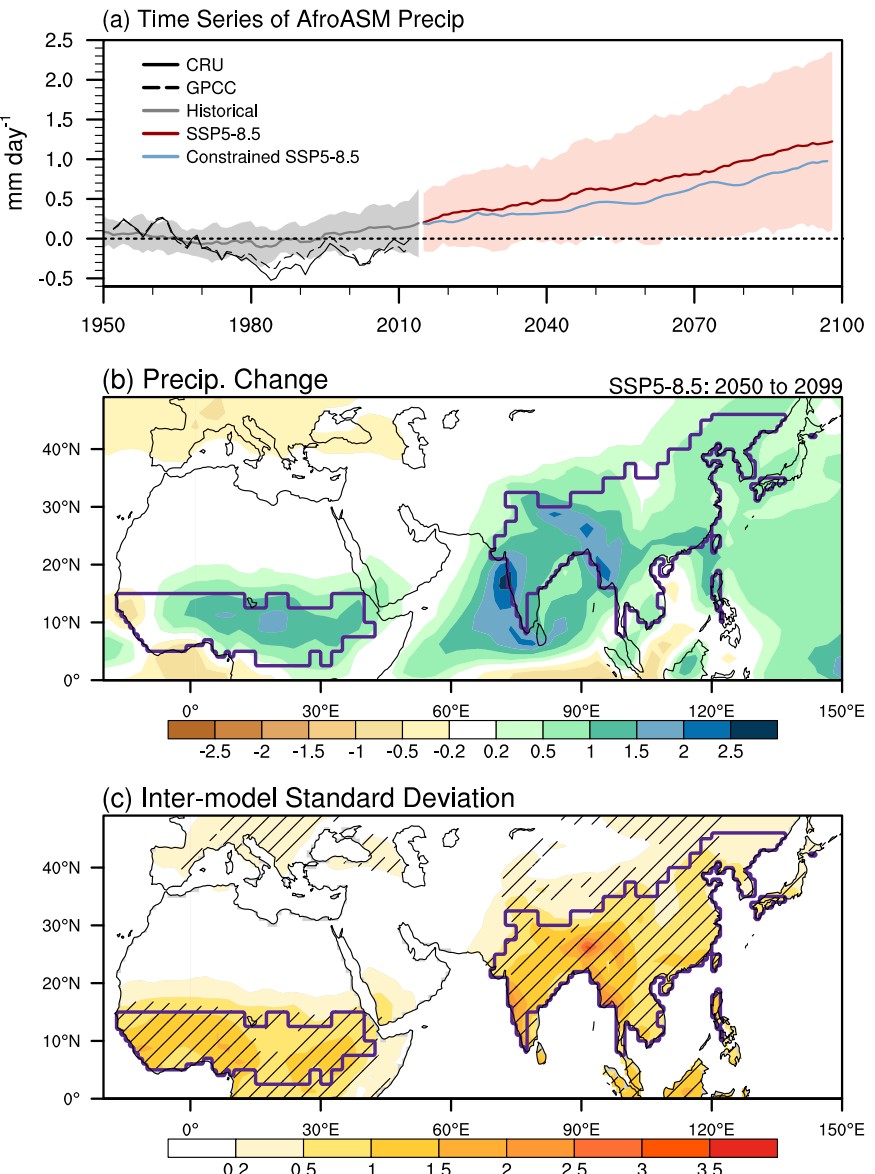

**Fig. 1 Projected changes in the Afro-Asian summer (June, July, August, and September) monsoon (AfroASM) precipitation and uncertainty of the projected changes. a** Time series of 5-year running mean of AfroASM precipitation anomalies (units: mm day$^{-1}$), relative to 1950~1980 mean. Historical (gray) and SSP5-8.5 (red) simulations are shown for the 5$^{th}$ and 95$^{th}$ percentiles across 30 models (shading), and the ensemble mean (thick solid lines). The blue solid line is the AfroASM precipitation anomalies after emergent constraint. The black solid and dash lines are the observational series from the Climatic Research Unit (CRU) Time-Series (TS) version 4.02 and Global Precipitation Climatology Centre version 7 (GPCC v7), respectively. **b** Changes in precipitation (units: mm day$^{-1}$) under SSP5-8.5 scenario (2050–2099) relative to historical simulation (1965–2014). The region surrounded by the contour is the Afro-Asian monsoon region (see Methods). **c** The inter-model standard deviation ($\sigma$) of projected precipitation changes. Hatched regions denote signal-to-noise ratio between the absolute value of projected changes and the standard deviation less than 1.5. The regions where precipitation changes are lower than 0.1 mm day$^{-1}$ or over ocean is omitted.

NH and the Arctic amplification[44–46]. Hence, the inter-model scatter of NH-SH ITC is associated with that of global mean warming rate under a specific radiative forcing, i.e., the equilibrium climate sensitivity (ECS) which measures warming magnitude under doubled $CO_2$ concentration relative to the pre-industrial period (Fig. 3a; see Methods). Models with a larger ECS show a larger NH-SH ITC in both the historical and future periods (Fig. 3b). A larger increase of ITC would lead to a stronger PC1 and thereby project more precipitation over AfroASM regions (Fig. 3c and Supplementary Fig. S1). The underlying physical mechanism is that a model with larger increase of ITC induces a stronger enhancement of low-level

cross-equatorial flow over North Atlantic Ocean, Somalia and South China Sea (Supplementary Figs. S3a and S3b). The pattern of low-level cross-equatorial flow regressed onto the PC1 across models closely resembles that regressed onto the projected ITC (Supplementary Figs. S3c and S3d). Hence a larger projected increase of ITC would induce a stronger low-level cross-equatorial flow and thereby more moisture transport, finally resulting in more increase of AfroASM precipitation.

The underlying mechanism provides a solid physical basis to the observational constraint. Thus, we can constrain the future inter-model uncertainty of AfroASM precipitation based on the present-day observed ITC trend.

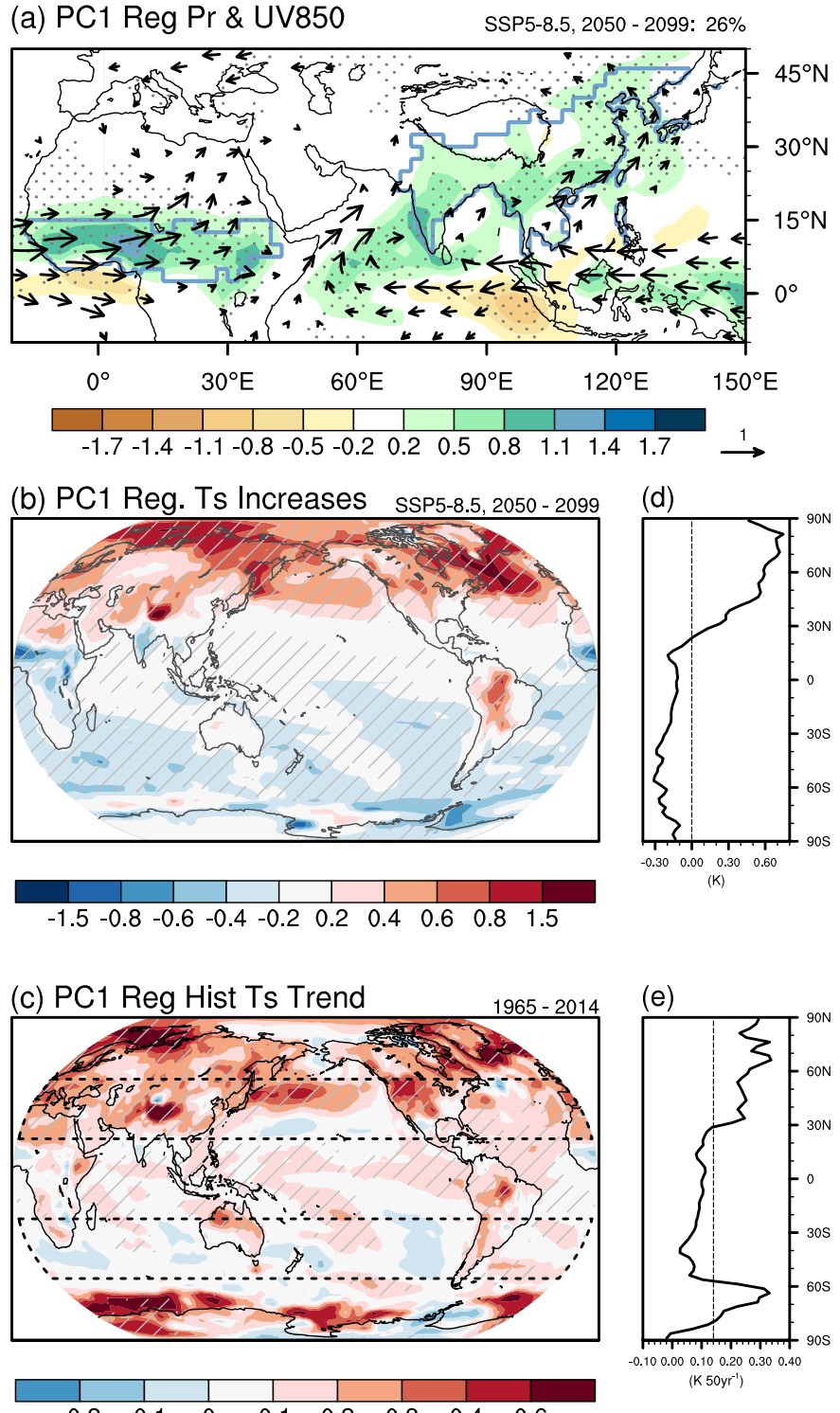

**Fig. 2 Dominant pattern of projected uncertainty and related historical pattern. a** The projected precipitation (shading, mm day$^{-1}$) and wind at 850 hPa (UV850; vector, m s$^{-1}$) across 30 CMIP6 models under high-emission scenario (SSP5-8.5) regresses onto the inter-model normalized leading principal components (PC1). The PC1 are derived from the inter-model empirical orthogonal function (EOF) analysis of projected precipitation change under SSP5-8.5 in 2050–2099 relative to 1965–2014 (see Methods). The percentage on the top-right corner is explained inter-model variance. **b** the future increase of surface temperature in 2050~2099 and **c** the trend of surface temperature (K) in 1965–2014 across models regresses onto the inter-model normalized PC1. Panels **d** and **e** are the zonal mean of the regression coefficient, and the thin dash vertical lines are the global area mean of the regression coefficient. The stippling, black vectors and hatching represent the regression exceeds 90% confidence level under Student's t test. Black dash boxes in **c** are used to define the pattern indices to constrain the PC1 (see Methods).

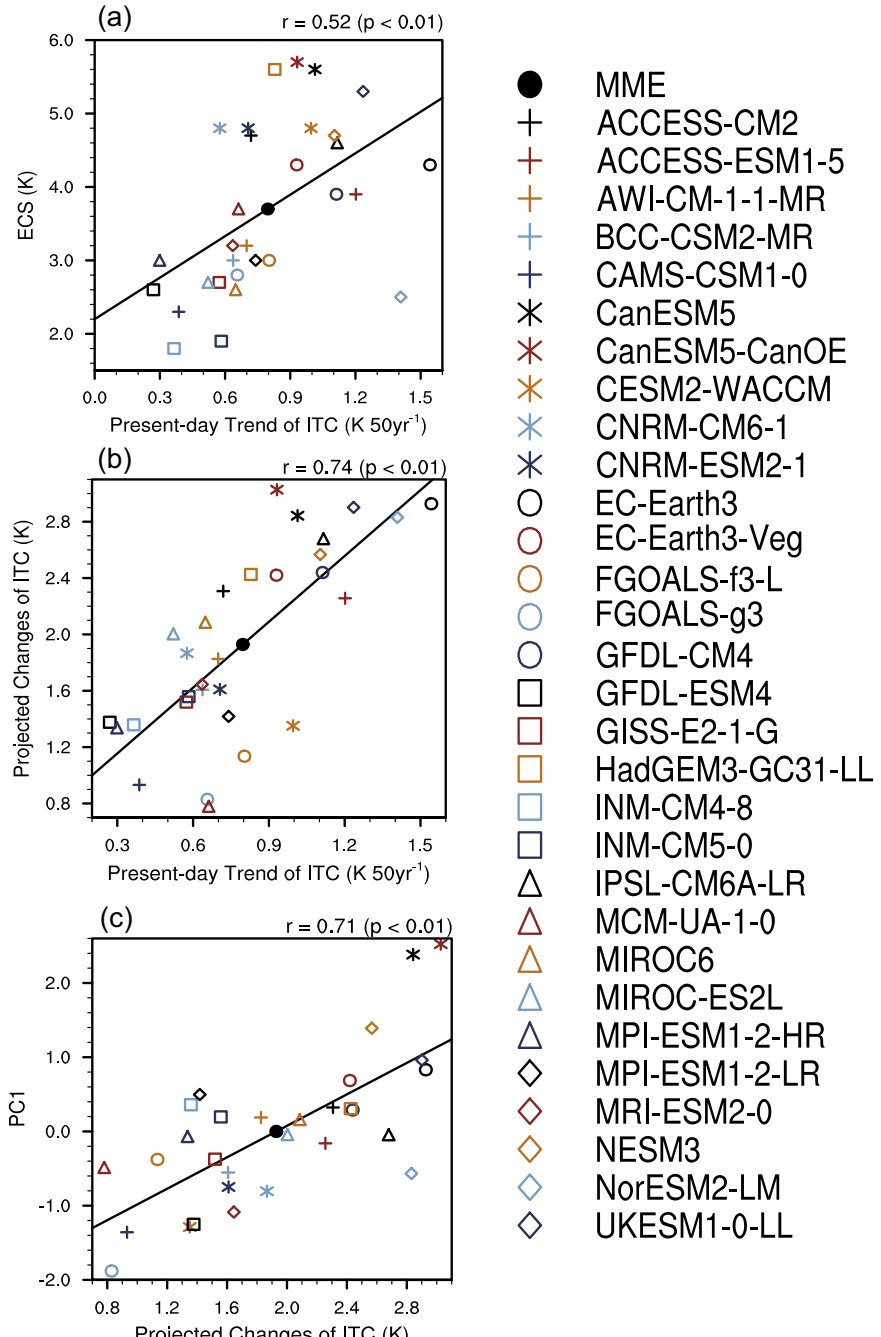

**Fig. 3 Inter-model physical relationship between the present-day and projected spread. a–c** inter-model relationship among the equilibrium climate sensitivity (ECS), present-day trend of interhemispheric thermal contrast (ITC), projected changes of ITC and PC1. The definition of the indices above is in Methods. Solid fitting line is obtained by the least square method. The results on the top-right corner are the correlation coefficient and significant level under Student's *t* test.

**Constrained projection of AfroASM precipitation**. The relationship between present-day warming patterns and projected spread across models allows the emergent constraint on the AfroASM precipitation, using multiple observation datasets. To measure the model's fidelity in simulating the present-day observed climate, we define an ITC pattern index ($ITC_I$). For each model, the $ITC_I$ is produced by projecting the present-day trend pattern of surface temperature onto the warming pattern associated with the inter-model PC1 shown in Fig. 2c. Using the above projecting pattern, the $ITC_I$ is defined as the difference between NH and SH (see Methods). The $ITC_I$ can well explain the leading mode of model uncertainty in projected AfroASM

precipitation, as evinced by the significant correlation coefficient with the area mean of AfroASM precipitation ($r = 0.58$, $p < 0.01$; Fig. 4a) and the PC1 ($r = 0.61$, $p < 0.01$; Fig. 4b), respectively.

Based on the relationship between present-day climate ($X$) and projected PC1 ($Y$), we constrain the PC1 using a linear fit: $Y = \bar{Y} + \rho(X - \bar{X})$, where $\rho$ is the corrected regression coefficient, and $\bar{X}$ and $\bar{Y}$ represent the present-day and projected multi-model ensemble (MME), respectively (see Methods). To constrain the PC1, we firstly calculate the mean $ITC_I$ based on four observational datasets (vertical red dashed line in Fig. 4). The values of $ITC_I$ simulated by ~70% models are larger than the observations, indicating a systematic overestimation of the

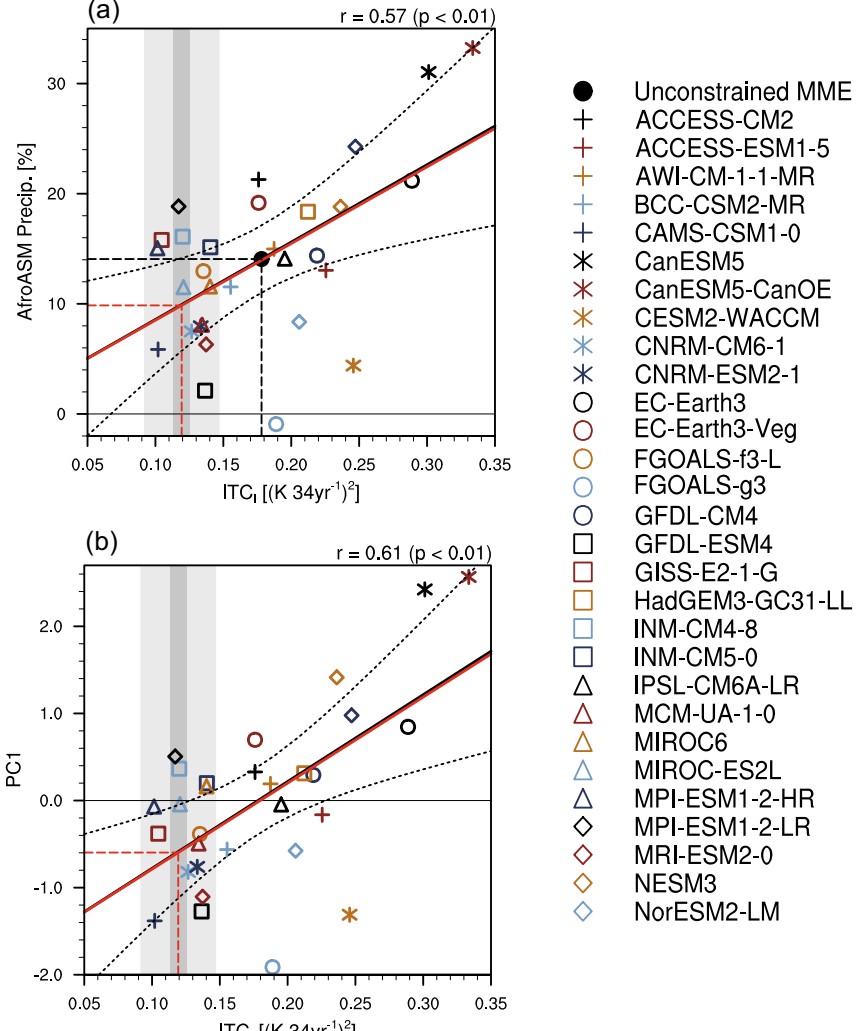

**Fig. 4 Relationship between spreads in projection of Afro-Asian summer monsoon (AfroASM) precipitation and historical warming pattern.** The scatter diagram between interhemispheric thermal contrast pattern index ($ITC_I$, (K 34 yr$^{-1}$)$^2$) across models in the present-day climate and inter-model spread of AfroASM precipitation (**a**) and normalized PC1 (**b**). $ITC_I$ can explain the PC1 with high corrected correlation coefficient (r) which is shown on the top right corner. Black fitting line is obtained by the least square method, and the red fitting line is an observational correction based on Eq. (5) (Eq. (5); see Methods). Dashed curves denote the 95% confidence range of the linear regression. The red (black) vertical and horizontal dash lines denote the mean of $ITC_I$ across multiple observation datasets (models) and the constrained (raw) projection, respectively. The dark gray shading denotes the range of ±1$\sigma$ across observation datasets. The light gray shading denotes the range contributed from the unforced internal variability (see Methods).

present-day ITC by CMIP6 models. The models with a larger ECS, such as CanESM5 and CanESM5-CanOE, simulate a stronger $ITC_I$ (Fig. 3a and Supplementary Table S3). The constrained AfroASM area-mean precipitation is $10 ± 6\%$ ($0.57 ± 0.38$ mm day$^{-1}$, constrained projection with ±1$\sigma$ across models after emergent constraint). The constrained value of PC1 is $-0.60 ± 0.80$.

Since the regional precipitation changes are informative and crucial for climate change adaptation activities, we further constrain the spatial patterns in the projection. Based on the observational constraint of PC1 value and PC1-related patterns, we attain the corrected projections of precipitation and circulation patterns (Fig. 5; see Methods). The constrained projection indicates an increase of AfroASM precipitation by 10% in 2050–099 relative to 1965–2014 (Fig. 5a), which is ~70% of that of the raw (viz, unconstrained) projection. Correspondingly, the constrained monsoon circulation change is weaker than the raw multi-model ensemble (Supplementary Fig. S4). Locally, the constrained increases over West African, East Asian, and South

Asian monsoon regions are 7%, 8%, and 12% respectively (Fig. 5a). The strongest reduction is seen over the West African monsoon region, where the constrained projection is only 49% of the raw MME projection, while over the East Asian monsoon region the constrained projection is 70% of the raw MME (Fig. 5b). The model uncertainty is also reduced after emergent constraint. Probability density function (PDF) of constrained PC1 is narrower than the original one, with a reduction of variance by 37% (Fig. 5c). Considering the explained variances of PC1 (26%), the total variance is reduced by ~10% (37% × 26%).

**Impacts on the potential water availability.** The AfroASM region holds a high density of population. More monsoon precipitation is expected to increase the potential water availability, which is mirrored in the runoff[47,48], while the associated intense monsoon precipitation will also lead to flood and landslide[49–52]. The projected increase in monsoon precipitation under global warming is expected to partly offset the drying tendency since the

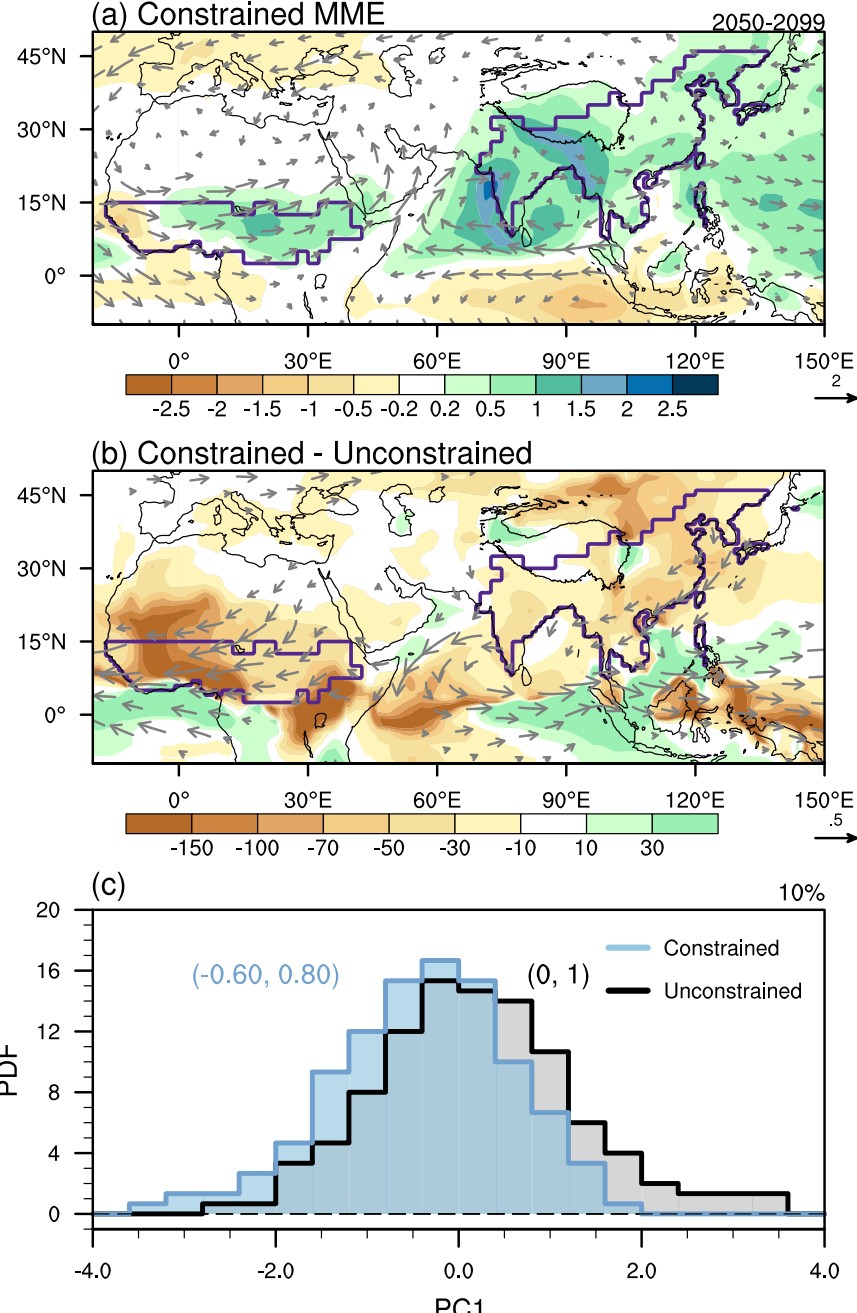

**Fig. 5 The constrained projection and narrowed uncertainty of Afro-Asian summer monsoon precipitation. a** The constrained precipitation (shading, mm day$^{-1}$) and wind at 850 hPa (UV850; vector, m s$^{-1}$; vectors drawn for larger than 0.1 m s$^{-1}$) based on the reconstruction of observed PC1, and (**b**) the constrained effect represented by the difference between constrained and unconstrained multi-model ensemble (MME). The constrained effect of precipitation in **b** is represented by the percentage (%) relative to the absolute values of unconstrained MME. Before calculating the fraction of constrained effect in **b**, the unconstrained MME are set as 0.05 mm day$^{-1}$ over the regions where the absolute values of unconstrained MME are lower than 0.05 mm day$^{-1}$. **c** Probability distribution function (PDF) of unconstrained (black) and constrained (blue) PC1. The values on the right corner are the narrowed variance due to emergent constraint. The values in the parenthesis are the mean and the standard deviation.

1950s (Fig. 1a). The smaller increase of precipitation in the constrained projection will reduce the increased potential water availability as expected from the raw projection, meanwhile the possible disasters related to heavy rainfall and floods will reduce accordingly. Here, we further estimate the impact of emergent constraint on the change in areal extent of precipitation and potential water availability.

To quantify the impact on the areal extent of precipitation, we examine the land area fraction that experiences a significant increase of precipitation (Fig. 6; see Methods). The fraction with a

significant increase of precipitation is 24% in the constrained projection, only 57% of the raw projection. Regionally, in the constrained projection, the land area fraction in the East Asian monsoon region is only 37% of the raw projection, while in the West African and South Asian monsoon regions, the corresponding results are 50% and 69%, respectively.

Based on the significantly positive correlation between precipitation and runoff (Supplementary Fig. S5), we further quantify the changes of potential water availability in the constrained projection (Fig. 6; see Methods). About 27% land

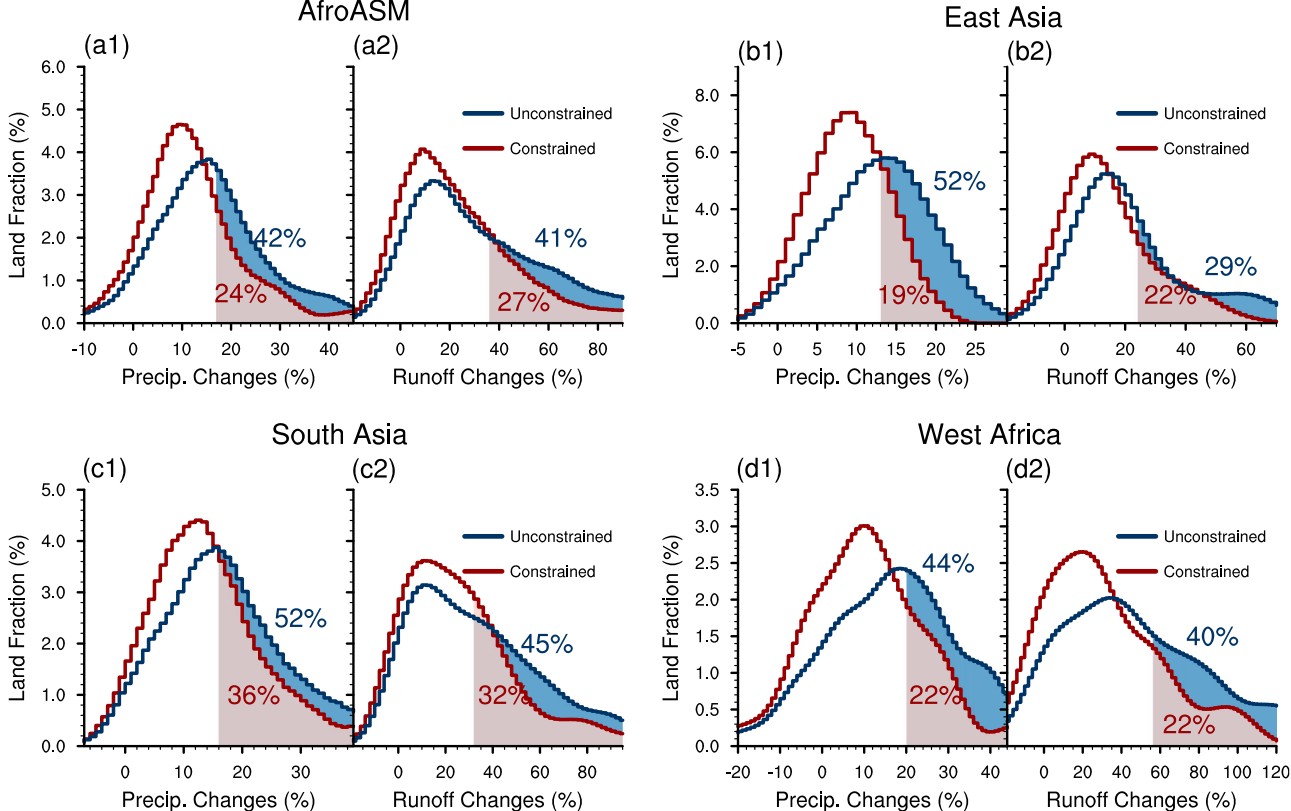

**Fig. 6 Spatial distribution of unconstrained and constrained changes in precipitation and runoff.** The probability distribution functions (PDFs) are aggregated by the land area fraction that experienced a certain change of precipitation (**a1–d1**) and runoff (**a2–d2**). The shadings and percentages in the subplots are the land area fraction which will experience a significant increase in the unconstrained (blue) and constrained (red) projections. The significant increase is defined as the increase exceeds the range of inter-model standard deviation (see Methods). The constrained projection of runoff is based on the relationship between runoff changes and precipitation changes (see Methods).

area in the AfroASM region will witness a significant increase of potential water availability in the constrained projection, which is only 66% of that of the raw projection. Regionally, the constrained land area fraction in the West African monsoon region is only 55% of the raw projection, while in the South Asia and East Asian monsoon regions, the corresponding result is 71% and 76%, respectively. Hence, less land area in AfroASM will experience a significant increase of precipitation and runoff in the constrained projection. These imply that the characteristics of precipitation change in the future will be milder than the raw projection.

## Discussion
Our emergent constraint on the projection of monsoon precipitation in the AfroASM regions is based on SSP5-8.5 scenario. We extend our analysis to the medium emission scenario of SSP2-4.5 and get similar conclusion (Supplementary Figs. S6 and S7). In addition, to examine the robustness of the emergent constraint, we check the inter-model correlation coefficient between $ITC_I$ and PC1 by using different subsets of model and including randomized outliers, and come to similar conclusion (Supplementary Fig. S8). The independence of the results on the model ensemble and the future emission scenarios confirms the robustness of the conclusions.

Our results reveal that the raw projection overestimates the increase of precipitation in the AfroASM region. While the constrained projection of the increase in AfroASM precipitation is 70% of the raw projection in the context of regional average, the effects of emergent constraint on the changes of precipitation and

water availability are more pronounced at regional scales. The projection of precipitation constrained by the observation is 49% (70%) of the raw projection in the West Africa (East Asia) monsoon region, even reduced by 70% (50%) more widely over sub-regional or local scales. The land fraction that will experience a significant increase of precipitation is 50% (37%) of that of the raw projection in West Africa (East Asia) monsoon region. The change of precipitation is echoed in the runoff as an indicator of potential water availability and the risk of flood. The smaller increase of potential water availability than the raw projection may pose a challenge to climate change adaptation and mitigation activities related to water management and food security[53,54], although a smaller than expected increase in rainfall will also reduce the risk of extreme precipitation and flooding.

Given that the inter-model uncertainty of global mean warming is closely associated with the inter-model spread of ECS[55,56], with normalizing by the global mean warming, the precipitation response (viz, hydrological sensitivity) still shows a remarkable spread across models[10,12,57]. A recent study reported that the projected uncertainty of hydrological sensitivity over the AfroASM regions is related to the projected uncertainty of ITC and land-sea thermal contrast[12]. Since the inter-model uncertainties of both hydrological sensitivity and global mean surface air temperature (GSAT) are significantly correlated with the ITC (Supplementary Fig. S9), we further constrain the hydrological sensitivity and GSAT separately based on the intermodel relationship between projected uncertainty and present-day biases (see Methods). The results based on constraining hydrological sensitivity and GSAT separately are consistent with that based on constraining the precipitation changes directly. The constraining

effects from hydrological sensitivity and GSAT reduce the raw precipitation projection by 13% and 21%, respectively. Thus, the GSAT warming plays a dominant role in the emergent constraint on precipitation changes, while the contribution from hydrological sensitivity should not be neglected.

## Methods

**Observations.** To correct the present-day model biases and constrain the future projections, multiple observational datasets are used. The monthly gridded observational surface temperature datasets used are (1) Berkeley Earth Surface Temperature (BEST)[58], (2) Cowtan and Way version 2[59], (3) NASA Goddard Institute for Space Studies Surface Temperature version 5 (GISTEMP v5)[60], and (4) NOAA Global Surface Temperature version 5.0 (NOAAGlobalTemp v5)[61,62]. The observational surface temperature datasets have been regridded on 2.5° × 2.5°.

Precipitation datasets are taken from (1) Climatic Research Unit (CRU) Time-Series version 4.02[63], (2) Global Precipitation Climatology Centre version 7 (GPCC v7)[64], (3) Climate Prediction Center (CPC) Merged Analysis of Precipitation (CMAP v1201)[65], and (4) the Global Precipitation Climatology Project (GPCP v2.3)[66]. Details of the datasets are provided in Supplementary Table 2.

**Model simulations.** We use the monthly output from 30 CMIP6 models (Supplementary Table S1) in historical simulation, and future projection under SSP5-8.5 scenario[35,37]. Given the recent trends in decarbonization, SSP5-8.5 is a highly unlikely scenario[38]. Hence, the monthly data from 18 CMIP6 models in future projection under median emission scenario, SSP2-4.5, are also used to verify the independence of emission scenarios. The first available realization for each model is used to give equal weight to each model. To examine the contribution from internal variability, we use the output of 29 CMIP6 models from the piControl simulations, in which all external forcings are held constant at their 1850 levels (Supplementary Table S1). All the data is re-gridded to 2.5° × 2.5° grids using first-order conservative interpolation, except for the circulation patterns which is re-gridded using bilinear interpolation.

To represent future projection of the summer monsoon rainfall, we focus on the changes in seasonal mean precipitation from June to September (JJAS) in 2050–2099 relative to the mean in 1965–2014.

**Afro-Asian monsoon.** The Afro-Asian monsoon region is defined as the land monsoon area over Eurasian continent and North Africa[3,67] (Fig. 1). The land monsoon domain is defined as the land area where the precipitation difference between the local summer and winter is larger than 2.0 mm day$^{-1}$, and local summer precipitation exceeds 55% of the annual total precipitation[67], based on the climatological mean of CMAP[65] and GPCP[66]. Local summer is defined as May to September for the Northern Hemisphere (NH). The Afro-Asian monsoon consists of 3 regional monsoons, including East Asia, South Asia, and North Africa monsoon (Fig. 1b, c contour).

A strong monsoon circulation is marked by a strengthening of the vertical zonal wind shear[68,69]. To quantify the response of AfroASM circulation, we define a circulation index as the vertical shear of zonal winds between 850 and 200 hPa averaged in a zone stretching from North Atlantic eastward to the Philippines (0–20°N, 30°W–120°E).

**Inter-model empirical orthogonal function (EOF) analysis.** The leading modes of inter-model uncertainty in the summer precipitation projection over Afro-Asian monsoon region is obtained by applying the typical EOF method to model-spatial dimension:

$$\triangle Pr'(m, n) \cong \sum_{i=1}^{num} \left( PC_{i,m} \times EOF_{i,n} \right),\qquad(1)$$

in which $\triangle$ denotes projected changes, $m$ denotes model number, $n$ denotes the spatial area, and $num$ is the mode number. Prime represents the deviation from the multi-model ensemble (MME). PCs are normalized here. The inter-model EOF method has been successfully used for the East Asia monsoon[29,31], tropical ocean SST[70–72], and extra-tropical oceans SST[73].

The leading principal component (PC1) of AfroASM precipitation accounts for 26% of the total intermodal variance (Fig. 2).

**Scaling individual models.** To confirm the results are not dependent on the strength of individual models' hydrological cycles, the mean precipitation changes over tropics and global for each model have been removed from the original precipitation changes, respectively, and then the precipitation changes have been normalized by the corresponding spatial standard deviation. We take the inter-model EOF analysis for the scaled precipitation changes. The results show that the patterns of projected changes of precipitation and low-level circulation regressed onto the scaled PC1 across models closely resemble that in Fig. 2a. Thus, we only present the results without scaling in the paper.

**The definition of interhemispheric thermal contrast (ITC) and ITC pattern index.** The future increase of AfroASM precipitation is closely associated with the projected NH-SH thermal contrast, with correlation coefficient higher than 0.7 across models[12,13,32]. To constrain the future projection using the present-day observation, we select two key regions which represent the NH-SH thermal contrast to define ITC and ITC pattern index ($ITC_I$).

The ITC is defined as the difference of the area-averaged surface temperature between (20°N~50°N, 0~360°) and (20°S~50°S, 0~360°). To represent the pattern of ITC, a pattern index is produced by projecting surface temperature trend onto the pattern associated with PC1 ($T_{PC1}$) shown in Fig. 2c.

The period 1981–2014 is chosen to calculate the pattern indices and constrain the projection. The warming trend of ITC in 1981–2014 has been dominated by the response to greenhouse gases, with no significant trend in aerosol cooling[74–78].

For the $ITC_I$ of each model, to clearly reflect the present-day warming pattern, the historical warming trend of surface temperature in each model ($T_{Hist}$) is projected onto the inter-hemisphere warming trend shown in Fig. 2c ($T_{PC1}$) in Northern Hemisphere (NH; 20°N~50°N, 0~360°) and Southern Hemisphere (SH; 20°S~50°S, 0~360°), respectively, following Chen[31]:

$$ITC_I = \left\langle T_{Hist} \cdot T_{PC1} \right\rangle_{NH} - \left\langle T_{Hist} \cdot T_{PC1} \right\rangle_{SH},\qquad(2)$$

where $\langle \rangle$ denotes area mean.

To calculate the index in observation, $T_{Hist}$ is derived from the four observational surface temperature datasets (Supplementary Table S2).

**Equilibrium climate sensitivity.** To investigate the source of model biases of the present-day ITC trend, we use the model's equilibrium climate sensitivity (ECS). The ECS is represented by the effective climate sensitivity which is estimated by regressing the net top-of-atmosphere radiance against the global mean surface air temperature changes in the first 150 years of the $CO_2$ quadruples experiment[75,78–80]. The ECS of most models in this study is derived from the Table 2 in Meehl[80], except for that of CanESM5-CanOE which is derived from Swart[81], and FGOALS-g3 which is derived from Zhou[82] and Li[83].

**The contribution from the unforced internal variability.** To quantify the impact from the unforced internal variability to the $ITC_I$, we calculate the $ITC_I$ based on the piControl simulations of 29 CMIP6 models. We calculate the trend of randomly selected continuous 34-year period and repeat this process over 1000 times to obtain 1000 synthetic members from each model. The contribution of the internal variability is measure by the variance across different synthetic members ($\sigma^2_{internal}$), which is approximately $7.8 \times 10^{-4}$ K$^2$ 34 yr$^{-2}$, compared to $3.9 \times 10^{-3}$ K$^2$ 34 yr$^{-2}$ of present-day $ITC_I$ across 30 models ($\sigma^2_{intermodel}$). The contribution from the internal variability accounts for 20% based on the variance ratio ($\sigma^2_{internal}/\sigma^2_{intermodel}$). The range of unforced internal variability in Fig. 4 (light gray shading) is represented by $\pm 1\sigma$ across different synthetic members ($\sigma^2_{internal}$).

**Hierarchical statistical framework for emergent constraint.** To constrain the projected AfroASM precipitation, we use the hierarchical emergent constraint (HEC) framework proposed by Bowman[24]. The HEC framework accounts for both the correlation between future and present-day climate, and the precision in the observational datasets, compared with the classical emergent constraint[24,31].

In the HEC framework, we establish a link between future climate change ($Y$) and present-day climate ($X$) to constrain $Y$. The emergent constraint is based on the linear regression between $Y$ and $X$ obtained from climate models:

$$Y = \bar{Y} + \rho_i \left( X - \bar{X} \right),\qquad(3)$$

where $\rho_i$ is the regression coefficient. $\bar{X}$ and $\bar{Y}$ are the multi-model ensemble mean of $X$ and $Y$, respectively. $Y$ is AfroASM precipitation changes or PC1, and $X$ is $ITC_I$ in Eq. (2).

Since we use the observation in current climate ($X_O$) to constrain $Y$, the uncertainty in the observations should be considered. Under the Gaussian assumptions which relates the observations to current climate[24], the signal-noise ratio (SNR) in the observation is the ratio between the variance across models ($\sigma^2_X$) and observational datasets ($\sigma^2_O$):

$$SNR = \sigma^2_X / \sigma^2_O,\qquad(4)$$

The regression coefficient is multiplied by $\frac{1}{1+SNR^{-1}}$ to correct $\rho_i$[31]. If the SNR is much larger than 1 (SNR » 1), the effect of correction can be neglected. In our analysis, SNR of $ITC_I$ is 110.

Based on the Eqs. (3) and (4), the constrained results and variance of future climate change $\bar{Y}_C$ can be expressed as:

$$\bar{Y}_C = \bar{Y} + \rho \left( \bar{X}_O - \bar{X} \right),\qquad(5)$$

$$\sigma^2_{Y_C} = \left( 1 - r^2 \right) \sigma^2_Y,\qquad(6)$$

Where $\rho$ is the corrected regression coefficient, i.e., $\frac{\rho_i}{1+SNR^{-1}}$; $r$ is the corrected correlation coefficient between $X$ and $Y$, i.e., $\frac{r_i^2}{1+SNR^{-1}}$, and the $r_i$ is the original correlation coefficient between $X$ and $Y$.

Based on the Eq. (6), the relative variance reduction $(1 - \frac{\sigma^2_{Y,C}}{\sigma^2_Y})$ from the HEC framework is $\frac{r_I^2}{1+SNR^{-1}}$. The total reduced variance (TRV) after constraining the PC1 can be expressed as the weighting on the corresponding explained variances of PC1 (PCV1):

$$TRV = \frac{r^2_{PC1,ITC_I}}{1 + SNR^{-1}_{ITC_I}} PCV1, \tag{7}$$

To constrain the projection results of individual model, we correct the PC1 of individual model using the $\bar{X}_O$. Base on the relationship of emergent constraint, the constrained PC1 of each model ($Y_{m,C}$) can be expressed as:

$$Y_{m,C} = Y_m + \rho(\bar{X}_O - X_m), \tag{8}$$

where $X_m$ and $Y_m$ is the $ITC_I$ and the PC1 of each model, respectively. The inter-model standard deviation of the constrained PC1 is 0.80.

**Corrected multi-model mean projection.** The corrected PC1 are estimated by the emergent constraint using the observed $ITC_I$. Since the pattern scaling has been shown to work robustly for seasonal averages of precipitation[22], the projection of the AfroASM precipitation can be corrected based on EOF reconstruction, following Eq. (1):

$$\triangle Pr = \triangle \overline{Pr} + \triangle Pr' \approx \triangle \overline{Pr} + PC1_O \times Pr'_{PC1}, \tag{9}$$

where subscript "O" denotes the corrected PC1 constrained by observation, $\triangle \overline{Pr}$ represents the multi-model ensemble mean of precipitation, and $Pr'_{PC1}$ represents the PC1-related pattern of precipitation changes shown in Fig. 2a. The wind fields at 850 hPa can be corrected in a similar way of Eq. (9), but the $Pr'_{PC1}$ term need to be replaced by the regression coefficients related to PC1.

**Spatial aggregated probability density function.** To measure the area with a certain change, we calculate the spatial aggregated probability density function (PDF) in the projected changes of precipitation. According to the latitude-dependent area, the grid points falling in each bin of the PDF have been weighted. Hence, the spatial distribution is an aggregated of all grid area satisfying the conditional sampling. The PDF is derived from the nonparametric assessment of the PDF. The spatial PDF is proposed by Fischer[84] and successfully used for the detection of extreme climate events[85–88].

**Land fraction that experiences a significant increase of precipitation.** The significant increase in the spatial aggregated PDF over AfroASM and three sub-monsoon regions is defined as the increase exceeds the inter-model standard deviation over AfroASM and three submonsoon regions, respectively. The area that experiences significant increases is aggregated spatially to represent the area which witness a significant increase of precipitation. Fraction is calculated with respect to the total area.

**Impact of precipitation change on runoff projection.** For each model, we calculate the projected changes in summer mean runoff for the period of 2050–2099 relative to the baseline. For the entire AfroASM and each submonsoon regions, there is a strong linear correlation between runoff changes and precipitation changes across models (Supplementary Fig. S5). This relationship enables a constraint on future runoff changes by using observationally constrained precipitation changes. Hence, following previous studies[48], the constrained changes of runoff ($\triangle R_{constrained}$) are derived as following equation:

$$\triangle R_{constrained} = k \cdot \triangle Pr_{constrained} + b, \tag{10}$$

where $\triangle Pr_{constrained}$ denotes the constrained change of precipitation based on Eq. (10), and $k$ denotes the regression coefficient between changes of runoff and precipitation, and $b$ denotes intercept.

**Constrained projection of hydrological sensitivity and GSAT.** The hydrological sensitivity is defined as the precipitation response normalized by the GSAT warming in 2050–2099 relative to 1965–2014. Since the inter-model spread of hydrological sensitivity is closely related to that of ITC (Supplementary Fig. S9a), which is consistent with recent study[12], we constrain the projection of hydrological sensitivity over AfroASM region based on Eq. (5) and Eq. (8). The constrained response of hydrological sensitivity is only 87% of that of the raw projection.

We constrained the projected GSAT warming using the observed GSAT trend in 1981–2014, following Tokarska[75] and Lee[45]. The constrained GSAT warming is $2.62 \pm 0.57$ K, weaker than the raw projection ($3.30 \pm 0.78$ K) under SSP5-8.5.

## Data availability
The data that support the findings of this study are freely available. CMIP6 model data are from the Earth System Grid Federation [https://esgf-node.llnl.gov/search/cmip6/]. Observational temperature BEST is from the Berkeley Earth [http://berkeleyearth.org/ data-new/], Cowtan and Way v2 is from the University of York and the University of Ottawa [https://www-users.york.ac.uk/~kdc3/papers/coverage2013/series.html], GISTEMP is from the NASA GISS [https://data.giss.nasa.gov/gistemp/]. Observational precipitation CRU is from the University of East Anglia [http://badc.nerc.ac.uk/data/cru/]. NOAAGlobalTemp v5, GPCC v7, CMAP v1201 and GPCP v2.2 are provided by the NOAA/OAR/ESRL, PSD, Boulder, CO, USA [https://psl.noaa.gov/data/gridded/].

## Code availability
The data in this study is analyzed with NCAR Command Language (NCL; http://www.ncl.ucar.edu/). The relevant codes in this work are available, upon request, from the corresponding author T. Z.

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

## Acknowledgements

This study is supported by the National Key Research and Development Program of China (2020YFA0608904), National Natural Science Foundation of China (Grant No. 41988101), and National Key Research and Development Program of China (2018YFA0606501). We thank the World Climate Research Programme's Working Group on Coupled Modeling, which is responsible for CMIP6, and the climate modeling groups (listed in Supplementary Table S1) for producing and making available their model output (https://esgf-node.llnl.gov/search/cmip6/). We also acknowledge NOAA/OAR/ESRL PSD for providing the observational precipitation datasets (Global Precipitation Climatology Project (GPCP v2.3): https://psl.noaa.gov/data/gridded/data.gpcp.html; Climate Prediction Center (CPC) Merged Analysis of Precipitation (CMAP v1201): https://psl.noaa.gov/data/gridded/data.cmap.html; Global Precipitation Climatology Centre version 7 (GPCC v7): https://www.esrl.noaa.gov/psd/data/gridded/data.gpcc.html) and the University of East Anglia Climatic Research Unit (CRU) for providing the observational precipitation datasets (https://catalogue.ceda.ac.uk/uuid/10d3e3640f004c578403419aac167d82).

## Author contributions

T.Z. designed the research, provided comments, and revised the manuscript. Z.C. performed the analysis and drafted the manuscript. X.C., W.Z., Lixia Z., Liwei Z. and M.W. provided comments, and helped to organize and revise the draft. All of the co-authors contributed to scientific interpretations and helped to improve the manuscript.

## Competing interests

The authors declare no competing interests.
