## [Peer Review File · Nature Communications]

REVIEWER COMMENTS

Reviewer #1 (Remarks to the Author):

Emergent constraint on future changes of summer precipitation in Afro-Asian monsoon regions finds less rich water resources than previous expectation

By Ziming Chen, Tianjun Zhou, Xiaolong Chen, Wenxia, Zhang, Lixia Zhang, Mingna Wu, and Liwei Zou

This study investigates the possibility to constraining future changes in precipitation in monsoon regions simulated by CMIP6 models. The models generally predict an increase in precipitation in these regions, yet with significant inter-model spread. This study aims to use the significant correlation between an observable metric (the inter-hemispheric thermal contrast) to constrain precipitation changes. The authors found that, given that since most models overestimate this thermal contrast, the observational constraint suggest a weaker moistening than would be expected from the CMIP model projections.

The article is interesting and the methodology is good and fairly well described. The approach used by the authors follows that of many papers that aim to constrain the uncertainty of different aspects of climate change simulated by CMIP models. That being said, the authors use a metric that has physical meaning, which allows for a physical understanding of the statistical inference.

I am less convinced by the part about the social impacts of the constrained projection (e.g. line 73). The authors simplify the issue of precipitation changes in monsoon regions. They defend the idea that a smaller than expected increase in rainfall is a bad thing for the population. This is not necessarily true, considering the possible disasters related to heavy rains, floods.... I would have liked more caution from the authors of the study.

Finally, I find the result relatively weak. While the emergent constraint (EC) constrains a relatively significant part of the PC1, it only constrains a significantly weaker part of the precipitation change. This occurs when your EC indirectly reduces the variable of interest. This reduces the interest of the study, even though the regional constraint part may seem to help improve it.

To summarize, the paper has no methodological problems, but I am only slightly convinced by the usefulness of the results.

Major comments:

- Line 29,168: The authors choose to highlight the increase in precipitation over land areas and extrapolate the impact on the population. However, I think this is a dangerous shortcut as most of the rainfall over monsoon regions is extreme and can cause flooding. Therefore, a smaller increase in monsoon precipitation may be better for the population in these regions than a large increase in extremes.**
- Line 157: The African monsoon region is not in North Africa. It is over the sub-saharian band.**
- Line 185-187: Changing the emission scenario does not prove the robustness of an EC. It only suggests that long-term CO2-induced global warming (and not natural variability) drive precipitation changes. Independence is complicated to obtain with EC. One way to do it would be to use a different set of models (subset of CMIP6 models, or using CMIP5 instead), but you would still have some inter-model similarity.**
- Line 188-189: It seems an interesting results, but it actually only reduced by 10% of the raw increase of precipitation increase.**

Minor comments:

- Title: 'Find'. Emergent constraints do not find results, but suggest. Overall, the title is ambiguous.
- Line 59-60: Not relevant, as several papers provide emergent constraints on different aspects of climate change.
- Line 73-80: Results should not be described in the introduction. This should be removed.
- Line 145: MME is not defined
- Line 202-203: I guess it is actually the HadCRUT4 dataset with Cowtan and Way modification?
- Line 219: Are observation data regridded on $2.5^\circ \times 2.5^\circ$ too?
- Lines 251-253: Results should not be in the method session
- Lines 263-265: Can the authors describe in more details how they calculate the
- Line 266: The ITC1 unit is not obvious. Could the authors explain it? It is related to the previous point.
- Figure 2: Please use the correct distance between latitudes, since the authors weight the ITC1 metric by area (i.e. the north pole is too highlighted).
- Figure 2: Is there any interesting information below 60°S ? Why do the authors remove this part of the globe?
- Line 646: "30 CMIP6"
- Figure 3a: ECS should be in the y-axis
- Figure 6a: I didn't understand how the figure is built. Is it the relative distribution an average of all model distribution or a aggregate of every points satisfying the conditional sampling for all models?
- Figure S1: Lines are confusing because models are not related. Please use dots for instance.

Reviewer #2 (Remarks to the Author):

The authors show that the intermodel spread in future Afro-Asian monsoon precipitation has a dominant mode of increased precipitation over land to the north, and reduced precipitation over the Equator to the south. They show this mode is related to the difference in inter-hemispheric temperature contrast between models.

As the modelled present-day and future inter-hemispheric thermal contrast are correlated, they argue that present-day inter-hemispheric thermal contrast can be used to constrain the future change in precipitation over the Afro-Asian monsoon region. They show that constraining projections in this way suggests less change in precipitation by the end of the century than the raw simulation data indicates.

Identifying sources of dynamical uncertainty in future monsoon precipitation is an ongoing challenge. Previous work, which the authors cite, has identified that interhemispheric and land-sea thermal contrast link to the spread in CMIP6 simulations (Wang et al. 2020) but did not propose a constraint as is done here. This study identifies a physically grounded emergent constraint on intermodel spread in projected rainfall, and so marks a valuable contribution to understanding future tropical rainfall changes. I am pleased to recommend this study for publication once some concerns with the methodology and framing are addressed.

Major comments

EOF analysis:

The physical arguments relating the increase Afro-Asian monsoon precipitation seen in Fig. 2a to the ITC seem sound, but I am concerned that a similar pattern might result simply from some models precipitating more strongly than others due to differences in their convection schemes. This would undermine the causality you base your emergent constrain on.

Did you explore whether scaling individual models to account for this prior to taking the EOF affected the results? e.g. by removing a mean (over the tropics or globally) for each model and normalize by the corresponding spatial standard deviation?

Robustness of constraint:

How sensitive is your relationship between PC1 and ITC1 to individual model datapoints? From Fig. 4 it seems the particularly high-PC1/high-ITC1 CanESM2 datapoints are key in giving a positive correlation. If you remove these datapoints, or remove some other randomized selection of models, are your constrained projections altered? Would the addition of an outlier considerably alter your results?

Framing:

You discuss 'Societal impacts of less increase in precipitation' from line 162, and refer to this point again on lines 193-194. You suggest that your findings are concerning because a smaller increase in precipitation will be a problem for agriculture in the Afro-Asian monsoon region. Is it actually desirable to have a greater increase in precipitation, and specifically, would it be problematic to have a smaller increase than the raw projections predict? You cite Schewe et al. 2014 in the conclusions to support the water scarcity issue, but that study looks at precipitation change globally. While their Fig. 1 indicates reduced annual discharge at 2 degrees warming across the Americas and Europe, it is not clear from their figures or discussion that the Afro-Asian monsoon region overall is expected to suffer water scarcity, besides perhaps China. While decreased precipitation can reduce water available for agriculture, increased precipitation can itself connect to flooding, loss of crops, increased occurrence of gastrointestinal and vector borne diseases. The raw projection in Fig. 1a suggests that by the end of the 21st Century we might on average see a 0.75mm/day increase in precipitation compared to 1950. If this is reduced in your constrained projection, might that not be a good thing? I don't think the reduction you predict is so great as to maintain precipitation below 1950s levels? I suggest that if you want to contextualize your results in this way, you should: Support your discussion on line 162 onwards with references to recent literature. Add a line to Fig. 1a to show the constrained precipitation projection.

Minor comments

Use of percentages: I found phrasing such as '26% less increase' (line 77) confusing. Does this mean the precipitation increase of the constrained projection is 74% of that of the raw projections? It would help to be explicit how you're comparing the constrained and raw precipitation somewhere in the main text.

Line 65: You discuss coherent rainfall variability 'on various timescales' across North Africa and Asia. Although this is true on millennial timescales, on shorter timescales the picture is less clear, and your phrasing doesn't specify what timescales you refer to. I suggest adjusting this line to separately cite the work looking at millennial and decadal timescales.

Line 84: SSP5-8.5 - acronym should probably be expanded? And perhaps it's worth stating this is a severe scenario for anyone not familiar with the CMIP6 experimental design.

Line 88-89: Clarify that 'ensemble mean and inter-model standard deviation' refer to the projected change.

Line 149: You could discuss here which models/model families particularly overestimate ITC1.

Line 164-165: "a remarkable drying tendency is seen in the observations in the latter half of the 20th century" - perhaps specifically guide the reader to compare rainfall in the 1950s with 1980s. Because of where the 0mm/day line sits, at first glance the weakening of precipitation in the 1980s did not look large to me.

Typos/grammar

Line 18: Afro-Asian summer monsoon -> The Afro-Asian summer monsoon
Line 18: populations -> people
Line 25: larger trend -> a larger trend
Line 25: more increase -> a greater increase
Line 26: present-day ITC trend -> the present-day ITC trend
Line 27: emergent constraint -> this emergent constraint
Line 29: significant increase -> a significant increase
Line 33: global monsoon system -> the global monsoon system
Line 34: Sahel -> the Sahel
Line 34: Indian -> the Indian
Line 37: populations -> people
Line 38: large spread -> a large spread
Line 54: present day -> the present day
Line 54: future -> the future
Line 79: fractions -> fraction
Line 85: western African monsoon -> the western African monsoon
Line 86: increases -> increase
Line 89: 90% Afro-Asian monsoon -> 90% of the Afro-Asian monsoon
Line 96: systematically -> systematic
Line 103: seasonal swing -> the seasonal swing
Line 125: Somali -> Somalia, or refer to enhancement of the Somali jet
Line 127: Hence a larger increase of ITC in the projection... -> Hence a larger projected increase of ITC...
Line 164: remarkably -> remarkable
Line 305: "The SNR is multiplied by" - should this be "the regression coefficient is multiplied by"?
Line 307: constraint -> constrained

Reviewer #3 (Remarks to the Author):

Summary

=====

The authors analyze 21st century changes in precipitation over land across the African and Asian monsoon sectors in CMIP6 models under a high-emissions scenario and attempt to develop an emergent constraint thereof based on the interhemispheric thermal contrast. They argue that this constraint narrows uncertainty and results in a mean projection of increased precipitation over these regions, but less of an increase than without the emergent constraint.

There are several interesting results here, but the manuscript has two critical problems that would need to be fixed before being a plausible candidate for publication: (1) its lack of disentangling the effect of ECS on the precip changes of interest, and (2) confusing and potentially problematic methods employed to generate the emergent constraint. These and more minor comments are detailed below.

Major comments

=====

Role of ECS

~~~~~

The manuscript notes that the projected summer rainfall changes of interest are well correlated with model equilibrium climate sensitivity (ECS). This is not entirely surprising, as it is well established that more generally hydrological cycle changes to first order scale with the global-mean warming. Held and Soden 2006 and Allen and Ingram 2002 are early and canonical

references here, but there is a whole literature on "pattern scaling" that discusses this in more detail, e.g. [tebaldi\_pattern\_2014].

Given that, as is standard in analyses of precip change under mean temperature change (I can't think of an example reference here, but there are plenty), the precip changes need to be normalized by the global-mean warming. This removes the direct influence of ECS and results in a clean disentangling of two presumably independent effects: (1) how much warming there is, and (2) given the amount of warming, how the monsoon precip will change. One can then address each factor separately, with constraints on ECS (e.g. very exhaustively addressed by [sherwood\_assessment\_2020]) and separately constraints such as what the present study attempts on the precip change per unit warming.

In other words, without normalizing by mean warming in each model, the present manuscript's attempt nominally to constrain Afro-Asian monsoon precip change implicitly ends up being a muddled attempt to constrain both ECS and hydrological sensitivity over the Afro-Asian monsoon region.

#### Complicated or confusing procedures on the emergent constraint

~~~~~

There are several steps of slicing and dicing of the data before ultimately arriving at the "ITC pattern index" that is used as the emergent constraint. Perhaps these procedures are all warranted, but as written I found them confusing and as such not entirely convincing as I'll now describe.

Most importantly, the "headline" figure is Figure 4, from which the quoted values of constrained projections in the abstract are derived. But what's being "constrained" isn't actually the projected precip change in the Afro-Asian monsoon sector---it's the PC1. The behavior of the PC itself isn't ultimately what we care about---what we care about is the actual change in rainfall over the region. But that's never shown if I'm understanding correctly; it's inferred from the constraint on the PC and its relationship to the PC.

Separately, I'm not sure that the uncertainties are being properly propagated forward into the constraint. Fig. 4 nicely includes measures of uncertainty on both the ITC index (grey shading) and in the regression line (dashed curves). In L149 the quoted "constrained" PC1 value is -0.49 ± 0.63 . But these error bounds surely underestimate the uncertainty (also it should be stated more clearly in the main text what specifically the ± 0.63 bounds are and how they were computed). I know this isn't a rigorous way of doing it, but just by eye examine in Fig. 4 the area contained within the two dashed curves and the grey shading as a plausible estimate of the uncertainty in both terms. Within that area, the intersection of the "obs" (vertical dashed line) with the projection (horizontal dashed line) could yield PC1 as low as roughly -1.4 or as high as 0.2 , a considerably wider range than -0.49 ± 0.63 . If I am mistaken and the uncertainty in both the observed ITC index *and* the model PC1-ITC1 regression are in fact both being accounted for, I do apologize, but ask that you clarify the description in the text. If I am not mistaken, then how much does this weaken your constraint?

In addition, L137-139 describe the ITC index used as the constraint as "produced by projecting the present-day trend pattern of surface temperature onto the warming pattern associated with the inter-model PC1 shown in Figure 2c". Why do this projection step and not simply use the present-day trend pattern itself? Your procedure may indeed be the superior one, but it needs to be justified in the manuscript.

Minor comments

=====

L2

"less rich"

L33-34

An ITCZ-like narrow band is a reasonable description over Africa but certainly not for the Indian and E. Asian monsoons, which are more complicated

L38-39

"raising risk on failure in addressing regional climate change" confusing wording; rephrase

L53

To me, the definitive citation on emergent constraints and how to do them in the most physically justifiable way is [klein_emergent_2015]. I recommend you cite it and ensure your approach checks all of their requirements for a sound constraint.

L69-70

I don't think the reference to the Scenario Model Intercomparison Project is necessary; just referring to CMIP6 is sufficient and has less danger of being confusing

L84

The manuscript should briefly note that SSP8.5 is now a highly unlikely scenario given recent trends in decarbonization; e.g. [hausfather_emissions_2020].

L96

systematically -> systematic

L102-104

The manuscript should briefly note that, even if the thermal contrast proves to be a useful predictor, modern understanding of the monsoon replaces thermal gradients with moist entropy or moist static energy gradients. The Geen et al review paper cited notes this; see also e.g. [hill_theories_2019].

L110-112

"rooted in" is too vague; please clarify. Also, from Fig. 2c there is much more noise for the historical, such that your claim is not immediately valid by eye from that plot. I recommend adding panels of zonal averages for both the future and historical to make this clearer.

L115-116

Should cite one or more papers of the voluminous literature on transient warming, Arctic amplification, etc.

L123-125

These are far removed from the E. Asian monsoon sector which you otherwise are including. In addition, similar to the suggestion above for Fig. 2 I recommend adding panels of zonal-mean fields: this is quite noisy to the point that I don't find the claim about clear increases in cross-equatorial flow totally convincing.

Fig. 4

Omit the "(a)" label since there's only one panel.

L250-251

I'm not familiar with these papers, but is this saying that the connection between the Afro-Asian monsoon precip change and the NH-SH thermal contrast has already been well established? If so, that needs to be stated in the main text, not just the methods, and it needs to be clarified specifically what in this study is novel (presumably all the emergent constraint material).

L256-259

Why these latitude ranges? Why do they differ between the hemispheres? Without more physical justification, this feels a little fishy.

L262

Can you really claim that internal variability influence is weak over a ~3 decade timescale on regional scales? C.f. Clara Deser and Karen McKinnon's work (among others) with large ensembles and internal variability.

Bibliography

=====

[tebaldi_pattern_2014] Tebaldi \& Arblaster, Pattern Scaling: *Strengths and Limitations, and an Update on the Latest Model Simulations*, *Climatic Change*, 122(3), 459--471 (2014). link. [doi](http://dx.doi.org/10.1007/s10584-013-1032-9).

[sherwood_assessment_2020] Sherwood, Webb, Annan, Armour, Forster, Hargreaves, Hegerl, Klein, Marvel, Rohling, Watanabe, Andrews, Braconnot, Bretherton, Foster, Hausfather, von der Heydt, Knutti, Mauritsen, Norris, Proistosescu, Rugenstein, Schmidt, Tokarska \& Zelinka, An Assessment of *Earth's* Climate Sensitivity Using Multiple Lines of Evidence, *Reviews of Geophysics*, n/a(n/a), e2019RG000678 (2020). link. [doi](http://dx.doi.org/10.1029/2019RG000678).

[klein_emergent_2015] Klein \& Hall, Emergent *Constraints* for *Cloud Feedbacks*, *Curr Clim Change Rep*, 1(4), 276--287 (2015). link. [doi](http://dx.doi.org/10.1007/s40641-015-0027-1).

[hausfather_emissions_2020] Hausfather \& Peters, Emissions *the 'Business as Usual' Story Is Misleading*, *Nature*, 577(7792), 618--620 (2020). link. [doi](http://dx.doi.org/10.1038/d41586-020-00177-3).

[hill_theories_2019] Hill, Theories for Past and Future Monsoon Rainfall Changes, *Curr Clim Change Rep*, 5(3), 160--171 (2019). link. [doi](http://dx.doi.org/10.1007/s40641-019-00137-8).

Response to reviewers' comments of NCOMMS-21-30260-T "Emergent constraints suggest overestimated future Afro-Asian monsoon precipitation increase"

We hope to express our appreciation to the reviewers for the constructive comments and suggestions, which have greatly helped us to improve the quality of the manuscript. Below, we have provided a point-by-point response to the comments. In the following, the reviewer's comments are written in black, followed by our response in blue.

Reviewer #1 (Remarks to the Author):

Emergent constraint on future changes of summer precipitation in Afro-Asian monsoon regions finds less rich water resources than previous expectation

By Ziming Chen, Tianjun Zhou, Xiaolong Chen, Wenxia, Zhang, Lixia Zhang, Mingna Wu, and Liwei Zou

This study investigates the possibility to constraining future changes in precipitation in monsoon regions simulated by CMIP6 models. The models generally predict an increase in precipitation in these regions, yet with significant inter-model spread. This study aims to use the significant correlation between an observable metric (the inter-hemispheric thermal contrast) to constrain precipitation changes. The authors found that, given that since most models overestimate this thermal contrast, the observational constraint suggest a weaker moistening than would be expected from the CMIP model projections.

The article is interesting and the methodology is good and fairly well described. The approach used by the authors follows that of many papers that aim to constrain the uncertainty of different aspects of climate change simulated by CMIP models. That being said, the authors use a metric that has physical meaning, which allows for a physical understanding of the statistical inference.

I am less convinced by the part about the social impacts of the constrained projection (e.g. line 73). The authors simplify the issue of precipitation changes in monsoon regions. They defend the idea that a smaller than expected increase in rainfall is a bad thing for the population. This is not necessarily true, considering the possible disasters related to heavy rains, floods.... I would have liked more caution from the authors of the study.

Finally, I find the result relatively weak. While the emergent constraint (EC) constrains a relatively significant part of the PC1, it only constrains a significantly weaker part of the precipitation change. This occurs when your EC indirectly reduces the variable of interest. This reduces the interest of the study, even though the regional constraint part may seem to help improve it.

To summarize, the paper has no methodological problems, but I am only slightly convinced by the usefulness of the results.

We would like to express our appreciation to the reviewer for the constructive comments that helped us to improve the manuscript. According to the reviewer's suggestions, we have thoroughly revised the manuscript. We hope that the new added results on the potential water availability and the regional effects of emergent constraint help to further convince readers of the usefulness of the results.

Major comments:

1. Line 29,168: The authors choose to highlight the increase in precipitation over land areas and extrapolate the impact on the population. However, I think this is a dangerous shortcut as most of the rainfall over monsoon regions is extreme and can cause flooding. Therefore, a smaller increase in monsoon precipitation may be better for the population in these regions than a large increase in extremes.

Thanks for your valuable comment. We agree that the factors affecting water resources include not only the precipitation amount, but also the characteristics of precipitation (Cho et al., 2016; Singh & Kumar, 2013; Li et al., 2016; Kundzewicz et al., 2019). Hence, in the revised manuscript, we have rephrased the statement and discussed the impact of precipitation change on the potential water availability, instead of highlighting the less increase of water resources as you suggested. We also stated that the increase of precipitation may cause negative impact such as flooding. For details, please see L28-L32, L175-L200 and L219-L222 in the revised manuscript. We also list the revision of the text below for your reference:

L28-L32: "The land area that will experience a significant increase of precipitation (runoff) is ~57% (66%) of that of the raw projection. A smaller increase of precipitation than raw projection will reduce the risk of extreme precipitation and flooding, while it may also pose a challenge to adaptation and mitigation activities related to water resources management."

L175-L200:

"Impact on the potential water availability

The AfroASM region holds a high density of population. More monsoon precipitation is expected to increase the potential water availability, which is mirrored in the runoff^{42,43}, while the associated intense monsoon precipitation will also lead to flood and landslide⁴⁴⁻⁴⁷. The projected increase in monsoon precipitation under global warming is expected to partly offset the drying tendency since the 1950s (Fig. 1a). The less increase of precipitation in the constrained projection will reduce the increased potential water availability as expected from the raw projection, meanwhile the possible disasters related to heavy rainfall and floods will reduce accordingly. Here, we further estimate the

impact of emergent constraint on the change in areal extent of precipitation and potential water availability.

To quantify the impact on the areal extent of precipitation, we examine the land area fraction that experiences a significant increase of precipitation (Fig. 6; see Methods). The fraction with a significant increase of precipitation is 24% in the constrained projection, only 57% of the raw projection. Regionally, in the constrained projection, the land area fraction in East Asian monsoon region is only 37% of the raw projection, while in West African and South Asian monsoon regions, the corresponding results are 50% and 69%, respectively.

Based on the significant positive correlation between precipitation and runoff (Supplementary Fig. S5), we further quantify the changes of potential water availability in the constrained projection (Fig. 6; see Methods). About 27% land area in the AfroASM region will witness a significant increase of potential water availability in the constrained projection, which is only 66% of that of the raw projection. Regionally, the constrained land area fraction in West African monsoon region is only 55% of the raw projection, while in South Asia and East Asian monsoon regions, the corresponding result is 71% and 76%, respectively. Hence, less land area in AfroASM will experience a significant increase of precipitation and runoff in the constrained projection. These imply that the characteristics of precipitation change in the future will be milder than the raw projection.”

L219-L222:

“The less increase of potential water availability than the raw projection may pose a challenge to climate change adaptation and mitigation activities related to water management and food security^{48,49}, although a smaller than expected increase in rainfall will also reduce the risk of extreme precipitation and flooding.”

Reference:

Cho, C., Li, R., Wang, S. Y., Yoon, J. H. & Gillies, R. R. Anthropogenic footprint of climate change in the June 2013 northern India flood. *Clim. Dyn.* **46**, 797–805 , doi: 10.1007/s00382-015-2613-2 (2016).

Kundzewicz, Z. W. et al. Flood risk and its reduction in China. *Advances in Water Resources* 130, 37–45, doi: 10.1016/j.advwatres.2019.05.020 (2019).

Li, C., Chai, Y., Yang, L. & Li, H. Spatio-temporal distribution of flood disasters and analysis of influencing factors in Africa. *Natural Hazards* 82, 721–731, doi: 10.1007/s11069-016-2181-8 (2016).

Singh, O. & Kumar, M. Flood events, fatalities and damages in India from 1978 to

2006. *Natural Hazards* 69, 1815–1834, doi: 10.1007/s11069-013-0781-0 (2013).

2. Line 157: The African monsoon region is not in North Africa. It is over the sub-saharian band.

Thanks. We have replaced the “North Africa” with “West Africa” or “West African monsoon region” in the revised manuscript, following the “Annex V: Monsoons” in the IPCC AR6 (2021). Please see L166-L168 in the revised manuscript:

Reference:

IPCC, 2021: Annex V: Monsoons [Cherchi, A., A. Turner (eds.)]. In: *Climate Change 2021: The Physical Science Basis. Contribution of Working Group I to the Sixth Assessment Report of the Intergovernmental Panel on Climate Change* [Masson-Delmotte, V., P. Zhai, A. Pirani, S. L. Connors, C. Péan, S. Berger, N. Caud, Y. Chen, L. Goldfarb, M. I. Gomis, M. Huang, K. Leitzell, E. Lonnoy, J.B.R. Matthews, T. K. Maycock, T. Waterfield, O. Yelekçi, R. Yu and B. Zhou (eds.)]. Cambridge University Press. In Press

3. Line 185-187: Changing the emission scenario does not prove the robustness of an EC. It only suggests that long-term CO₂-induced global warming (and not natural variability) drive precipitation changes. Independence is complicated to obtain with EC. One way to do it would be to use a different set of models (subset of CMIP6 models, or using CMIP5 instead), but you would still have some inter-model similarity.

Thanks for your suggestion. As suggested, we use different model ensembles to verify the robustness of the emergent constraint. Firstly, we deduct one to four models from 30 CMIP6 models in sequence, and we get 30, 435, 4060 and 27405 subsets of model ensemble, respectively. For instance, there are 435 subsets of model ensemble when we deduct two models from the raw 30-model ensemble ($C_{30}^2 = 435$). Then, we calculate the correlation coefficient between ITC_I and PC1 in each subset, and get correlation coefficients of 0.61 (0.56~0.67 for the 5th-95th range), 0.61 (0.53~0.69), 0.61 (0.51~0.71) and 0.61 (0.49~0.72) by deducting one, two, three and four models, respectively. The emergent constraint in all these subsets is statistically significant, as all the ranges of correlation coefficient exceed the 5% significant level using student t-test.

Hence, the established emergent constraint is independent of the model ensembles, confirming the robustness of the conclusion. Please see L204-L208 in the revised manuscript, and Supplementary Note #1 and Figure S8 in the Supplementary Information. In addition, since the SSP scenarios include regional forcings which are different from the CMIP5 RCPs, and the model response and the main source of the inter-model spread of CMIP6 may be different from that of CMIP5, comparability between CMIP5 and CMIP6 scenarios cannot be established for detailed assessments (Meinshausen et al., 2020; Wyser et al., 2020; Lee et al., 2021). So we only use the

CMIP6 model ensemble here.

We also list the revised text below for your reference:

L204-L208: “In addition, to examine the robustness of the emergent constraint, we check the inter-model correlation coefficient between ITC_I and PC1 using different model ensembles, and come to similar conclusion (Supplementary Fig. S8). The independence of the results on the model ensemble and the future emission scenarios confirms the robustness of the conclusions.”

The information added in the Supplementary Information is also listed below:

L73-L88: “**Supplementary Information Note #1:**

We use different model ensembles to verify the robustness of the emergent constraint. Firstly, we deduct one to four models from the 30 CMIP6 models in sequence, and we get 30 ($C_{30}^1 = 30$), 435 ($C_{30}^2 = 435$), 4060 ($C_{30}^3 = 4060$) and 27405 ($C_{30}^4 = 27405$) subsets of model ensemble, respectively. Then, we calculate the relationship between ITC_I and PC1 in each subset. The results are shown as the blue bars in Supplementary Figure S8.

To exam whether the addition of outliers would alter the relationship of emergent constraint, we randomly add one to four outliers to the raw 30-model ensemble (“+1” to “+4” of the red bars in the Supplementary Figure S8). The outliers are created randomly in the range of PC1 and ITC_I , respectively. We repeat the above processes 1000 times to form 1000 synthetic members. The results are shown as the red bars in Supplementary Figure S8.

The robustness is defined as the range of correlation coefficient in different subsets of model ensemble exceeding the thresholds of the 5% significant level under student t-test.”

Supplementary Figure S8. Robustness test for the emergent constraint using different sets of model ensembles. The relationship is represented by the inter-model correlation coefficient between interhemispheric thermal contrast pattern index (ITC_I) and normalized PC1. The bar charts and the error bars represent the ensemble mean and range of 5th -95th percentile of correlation coefficient across different set of models. The methods on how to create different sets of models is described in Supplementary Information Note #1. The red dash curve shows the thresholds of the 5% significant level under the student t-test.

References:

Meinshausen, M. et al. The shared socio-economic pathway (SSP) greenhouse gas concentrations and their extensions to 2500. *Geoscientific Model Development* 13, 2571–3605, doi: 10.5194/gmd-13-3571-2020 (2020).

Lee, J. Y. et al. Chapter 4: Future global climate: scenario-based projections and near-term information. in *Climate Change 2021: The Physical Science Basis. Contribution of Working Group I to the Sixth Assessment Report of the Intergovernmental Panel on Climate Change* (eds. Masson-Delmotte, V. et al.) (Cambridge University Press, 2021).

Wyser, K., Kjellström, E., Koenig, T., Martins, H. & Döscher, R. Warmer climate projections in EC-Earth3-Veg: The role of changes in the greenhouse gas concentrations from CMIP5 to CMIP6. *Environmental Research Letters* 15, 054020, doi: 10.1088/1748-9326/ab81c2 (2020).

4. Line 188-189: It seems an interesting results, but it actually only reduced by 10%

of the raw increase of precipitation increase.

Thanks. We have rephrased this statement as “While the constrained projection of the increase in AfroASM precipitation is ~70% of the raw projection in the context of regional average, the effects of emergent constraint on the changes of precipitation and water availability are more pronounced at regional scales.”

To highlight the effects of emergent constraint, we have clarified the regional impacts in the revised manuscript. Please see L213-L217 in the revised manuscript:

L213-L217: “The projection of precipitation constrained by the observation is 49% (70%) of the raw projection in West Africa (East Asia) monsoon region, indicating a reduction exceeding 70% (50%) over a large part of West African (East Asian) monsoon region. The land fraction that will experience a significant increase of precipitation is 50% (37%) of that of the raw projection in West Africa (East Asia) monsoon region.”

Minor comments:

1. Title: ‘Find’. Emergent constraints do not find results, but suggest. Overall, the title is ambiguous.

Thanks. We have revised the title as “Observationally constrained projection of Afro-Asian monsoon precipitation”.

2. Line 59-60: Not relevant, as several papers provide emergent constraints on different aspects of climate change.

Thanks. Deleted.

3. Line 73-80: Results should not be described in the introduction. This should be removed.

Thanks. Deleted.

4. Line 145: MME is not defined

Thanks. We have added the definition of MME. Please see L150-L151 in the revised manuscript:

5. Line 202-203: I guess it is actually the HadCRUT4 dataset with Cowtan and Way modification?

Thanks. The HadCRUT4 has been replaced with NOAA Global Surface Temperature version 5.0 (NOAAGlobalTemp v5; Vose et al., 2012; Zhang et al., 2019). The NOAAGlobalTemp dataset is a merged land-ocean surface temperature analysis with a spatial resolution of $5^\circ \times 5^\circ$. Please see L242-L243 and Supplementary Table S2 in

the revised manuscript:

References:

Vose, R. S. et al. NOAA's merged land-ocean surface temperature analysis. *Bulletin of the American Meteorological Society* 93, 1677–1685, doi: 10.1175/BAMS-D-11-00241.1 (2012).

Zhang, H.-M. et al. Updated temperature data give a sharper view of climate trends. *Eos* 100, doi: 10.1029/2019EO128229 (2019).

6. Line 219: Are observation data regridded on $2.5^\circ \times 2.5^\circ$ too?

Thanks. Yes. The monthly gridded observational surface temperature datasets are regridded on $2.5^\circ \times 2.5^\circ$. We also used the observational surface temperature datasets in the original resolution to calculate the interhemispheric thermal contrast index (ITC_I) and constrain the PC1. The constrained PC1 is -0.60 ± 0.80 , which is same as that regridded on $2.5^\circ \times 2.5^\circ$. We have clarified it in the revised manuscript. Please see L243-L244 in the revised manuscript:

7. Lines 251-253: Results should not be in the method session.

Thanks. Deleted.

8. Lines 263-265: Can the authors describe in more details how they calculate the ITC_I .

Thanks. The detailed description of ITC_I is added in the revised manuscript. For details, please L309-L316 in the revised manuscript:

9. Line 266: The ITC_I unit is not obvious. Could the authors explain it? It is related to the previous point.

Thanks. Since the units of T_{Hist} and T_{PC1} are $K \ 34yr^{-1}$, the units of ITC_I is $(K \ 34yr^{-1})^2$.

10. Figure 2: Please use the correct distance between latitudes, since the authors weight the ITC1 metric by area (i.e. the north pole is too highlighted).

Thanks. Corrected. Please see Figure 2 in the revised manuscript.

11. Figure 2: Is there any interesting information below 60° S? Why do the authors remove this part of the globe?

Thanks. We have extended the latitude range in Figure 2, and found a significant positive anomaly related to PC1 across models in present day below 60° S. The positive anomaly may be associated with the model biases of mixed-phase cloud (Trenberth & Fasullo, 2010; Lawson & Gettelman, 2014). We have added a statement

in the revised manuscript. Please see Figure 2 and L113-L115 in the revised manuscript.

L113-L115: “In addition, a remarkable warming anomaly related to PC1 in the historical period is seen over the Southern Ocean, which may be associated with the model biases of mixed-phase cloud^{36,37}.”

Figure 2. Dominant pattern of projected uncertainty and related historical pattern. (a) The projected precipitation (shading, mm day⁻¹) and wind at 850 hPa (UV850; vector, m s⁻¹) across 30 CMIP6 models under high-emission scenario (SSP5-8.5) regresses onto the inter-model normalized leading principal components (PC1). The PC1 are derived from the inter-model empirical orthogonal function (EOF) analysis of projected precipitation change under SSP5-8.5 in 2050-2099 relative to 1965-2014 (see Methods). The percentage on the top-right corner is explained inter-model

variance. (b) the future increase of surface temperature in 2050~2099 and (c) the trend of surface temperature (K) in 1965-2014 across models regresses onto the inter-model normalized PC1. The right panels in (b) and (c) are the zonal mean of the regression coefficient, and the thin dash vertical lines are the global area mean of the regression coefficient. The stippling, black vectors and hatching represent the regression exceeds 90% confidence level under student t-test. Black dash boxes in (c) are used to define the pattern indices to constrain the PC1 (see Methods).

Reference:

Lawson, R. P. & Gettelman, A. Impact of Antarctic mixed-phase clouds on climate. *Proc. Natl. Acad. Sci. U. S. A.* **111**, 18156–18161, doi: 10.1073/pnas.1418197111 (2014).

Trenberth, K. E. & Fasullo, J. T. Simulation of present-day and twenty-first-century energy budgets of the southern oceans. *J. Clim.* **23**, 440–454, doi: 10.1175/2009JCLI3152.1 (2010)

12. Line 646: "30 CMIP6"

Thanks. Corrected.

13. Figure 3a: ECS should be in the y-axis.

Thanks. Corrected.

14. Figure 6a: I didn't understand how the figure is built. Is it the relative distribution an average of all model distribution or an aggregate of every point satisfying the conditional sampling for all models?

Thanks. The spatial distribution in Figure 6 is an aggregate of all grid area satisfying the conditional sampling for the multi-model ensemble, following Fischer et al. (2013). To avoid misleading, we have described it with more details in L385-L388 in the revised manuscript. We also list the description below:

L385-L388: "According to the latitude-dependent area, the grid points falling in each bin of the PDF have been weighted. Hence, the spatial distribution is an aggregated of all grid area satisfying the conditional sampling. The PDF is derived from the nonparametric assessment of the PDF."

References:

Fischer EM, Beyerle U, Knutti R. Robust spatially aggregated projections of climate extremes. *Nature Climate Change* 3, 1033-1038, doi:10.1038/nclimate2051 (2013)

15. Figure S1: Lines are confusing because models are not related. Please use dots for

instance.

Thanks. Corrected.

Reviewer #2 (Remarks to the Author):

The authors show that the intermodel spread in future Afro-Asian monsoon precipitation has a dominant mode of increased precipitation over land to the north, and reduced precipitation over the Equator to the south. They show this mode is related to the difference in inter-hemispheric temperature contrast between models.

As the modelled present-day and future inter-hemispheric thermal contrast are correlated, they argue that present-day inter-hemispheric thermal contrast can be used to constrain the future change in precipitation over the Afro-Asian monsoon region. They show that constraining projections in this way suggests less change in precipitation by the end of the century than the raw simulation data indicates.

Identifying sources of dynamical uncertainty in future monsoon precipitation is an ongoing challenge. Previous work, which the authors cite, has identified that interhemispheric and land-sea thermal contrast link to the spread in CMIP6 simulations (Wang et al. 2020) but did not propose a constraint as is done here. This study identifies a physically grounded emergent constraint on intermodel spread in projected rainfall, and so marks a valuable contribution to understanding future tropical rainfall changes. I am pleased to recommend this study for publication once some concerns with the methodology and framing are addressed.

We would like to express our appreciation to the reviewer for the constructive comments that helped us to improve the manuscript. According to the reviewer's suggestions, we have thoroughly revised the manuscript. We hope the results from the scaling EOF analysis and significant relationship of emergent constraint help to further demonstrate the robustness of the conclusion.

Major comments

1. EOF analysis:

The physical arguments relating the increase Afro-Asian monsoon precipitation seen in Fig. 2a to the ITC seem sound, but I am concerned that a similar pattern might result simply from some models precipitating more strongly than others due to differences in their convection schemes. This would undermine the causality you base your emergent constrain on.

Did you explore whether scaling individual models to account for this prior to taking the EOF affected the results? e.g., by removing a mean (over the tropics or globally) for each model and normalize by the corresponding spatial standard deviation?

Thanks for your comments and suggestions. As suggested, to assess whether the projected uncertainty results from some models, we scale each model before taking the inter-model EOF analysis. To scale each model, we remove the mean precipitation over tropics and global from the projected changes of precipitation, respectively, and

then normalize the projected changes by the corresponding spatial standard deviation. The results are shown in Figure R1 below. We found that the patterns of projected changes of precipitation and low-level circulation regressed onto the scaled PC1 across models closely resemble the pattern in Figure 2a in the revised manuscript. Both of two PC1-related patterns exhibit a spatially consistent increase of precipitation over the AfroASM domain. In addition, the scaled PC1 accounts for 24% of inter-model variance (Fig. R1), which is close to that in the unscaled EOF analysis (Fig. 2a in the revised manuscript). Hence, the synchronized precipitation changes in the model spread are dominated by the inter-model uncertainty of ITC.

Based on your comments, we have added the following statements to clarify that the projected uncertainty does not result from some individual models (Please also see L94-L97 and L289-L296 in the revised manuscript)

L94-L97: “To exclude that the above pattern may be dominated by strong diversity in mean precipitation and spatial variability across model, we scale each model prior to taking the inter-model EOF analysis (see Methods), and obtain similar patterns compared with that in Figure 2a.”

L289-L296: “**Scaling individual models**

To scale each model, the mean precipitation changes over tropics and global for each model have been removed from the original precipitation changes, respectively, and then the precipitation changes have been normalized by the corresponding spatial standard deviation. We take the inter-model EOF analysis for the scaled precipitation changes. The results show that the patterns of projected changes of precipitation and low-level circulation regressed onto the scaled PC1 across models closely resemble that in Figure 2a. Thus, we only present the results without scaling.”

Figure R1. Same as the Figure 2a in the revised manuscript, but the precipitation changes in the projection is scaled before taking the inter-model empirical orthogonal function (EOF) analysis. To scale the precipitation changes in each model, the mean precipitation changes over tropics (a, 30S~30N) and global (b, 90S~90N) have been removed from the original precipitation changes, and then the precipitation changes have been normalized by the corresponding spatial standard deviation.

2. Robustness of constraint:

How sensitive is your relationship between PC1 and ITC1 to individual model datapoints? From Fig. 4 it seems the particularly high-PC1/high-ITC1 CanESM2 datapoints are key in giving a positive correlation. If you remove these datapoints, or remove some other randomized selection of models, are your constrained projections altered? Would the addition of an outlier considerably alter your results?

Thanks. To demonstrate the robustness of the emergent constraint, we select different model ensembles from 30 CMIP6 models, and then re-examine the relationship of the emergent constraint (blue bars in Supplementary Fig. S8). The results show that the relationship of the emergent constraint in all of members are statistically significant at 5% level under student t-test (Supplementary Fig. S8). In addition, as suggested, we have tested adding one to four additional outliers to 30 CMIP6 models' ensemble, and

then check the relationship of the emergent constraint (red bars in Supplementary Fig. S8). We repeat these processes over 1000 times. The relationship of the emergent constraint is statistically significant at 5% level under student t-test. Hence, the relationship between PC1 and ITC_1 is not sensitive to individual model datapoints.

Please see Supplementary Figure S8 and L204-L208 in the revised manuscript.

3. Framing:

You discuss “Societal impacts of less increase in precipitation” from line 162 and refer to this point again on lines 193-194. You suggest that your findings are concerning because a smaller increase in precipitation will be a problem for agriculture in the Afro-Asian monsoon region.

Is it actually desirable to have a greater increase in precipitation, and specifically, would it be problematic to have a smaller increase than the raw projections predict? You cite Schewe et al. 2014 in the conclusions to support the water scarcity issue, but that study looks at precipitation change globally. While their Fig. 1 indicates reduced annual discharge at 2 degrees warming across the Americas and Europe, it is not clear from their figures or discussion that the Afro-Asian monsoon region overall is expected to suffer water scarcity, besides perhaps China.

While decreased precipitation can reduce water available for agriculture, increased precipitation can itself connect to flooding, loss of crops, increased occurrence of gastrointestinal and vector borne diseases. The raw projection in Fig. 1a suggests that by the end of the 21st Century we might on average see a 0.75mm/day increase in precipitation compared to 1950. If this is reduced in your constrained projection, might that not be a good thing? I don't think the reduction you predict is so great as to maintain precipitation below 1950s levels?

I suggest that if you want to contextualize your results in this way, you should: Support your discussion on line 162 onwards with references to recent literature. Add a line to Fig. 1a to show the constrained precipitation projection.

Thanks. Reviewer #1 is also concerned about the implications of less increase in precipitation. We agree that the factors affecting water resources include not only the precipitation amount, but also the characteristics of precipitation (Singh & Kumar, 2013; Li et al., 2016a; Kundzewicz et al., 2019). As suggested, we have revised the discussion of the societal impact of the constrained precipitation and added the time series of the constrained precipitation in Figure 1a in the revised manuscript. We stated that the risk of extreme precipitation, flood and landslide will change consequently according to the recent studies (Kundzewicz et al., 2019; Cho et al., 2016; Ranasinghe et al., 2021). Given that the regional precipitation change is mirrored in runoff, affecting the risk of flood, we quantify the impact of constrained projection by examining the fraction of land area that experiences a significant increase of precipitation and runoff. The results show that the land area that will

experience a significant increase of precipitation and runoff in the constrained projection is only ~57% and ~66% of that of the original expectation. These imply that the characteristics of precipitation in the future will be milder than the raw projection. Please see L175-L200 and L219-L222 and Figure 1a in the revised manuscript. We also list the revised statements below for your reference:

L175-L200:

“Impacts on the potential water availability

The AfroASM region holds a high density of population. More monsoon precipitation is expected to increase the potential water availability, which is mirrored in the runoff^{42,43}, while the associated intense monsoon precipitation will also lead to flood and landslide⁴⁴⁻⁴⁷. The projected increase in monsoon precipitation under global warming is expected to partly offset the drying tendency since the 1950s (Fig. 1a). The less increase of precipitation in the constrained projection will reduce the increased potential water availability as expected from the raw projection, meanwhile the possible disasters related to heavy rainfall and floods will reduce accordingly. Here, we further estimate the impact of emergent constraint on the change in areal extent of precipitation and potential water availability.

To quantify the impact on the areal extent of precipitation, we examine the land area fraction that experiences a significant increase of precipitation (Fig. 6; see Methods). The fraction with a significant increase of precipitation is 24% in the constrained projection, only 57% of the raw projection. Regionally, in the constrained projection, the land area fraction in East Asian monsoon region is only 37% of the raw projection, while in West African and South Asian monsoon regions, the corresponding results are 50% and 69%, respectively.

Based on the significant positive correlation between precipitation and runoff (Supplementary Fig. S5), we further quantify the changes of potential water availability in the constrained projection (Fig. 6; see Methods). About 27% land area in the AfroASM region will witness a significant increase of potential water availability in the constrained projection, which is only 66% of that of the raw projection. Regionally, the constrained land area fraction in West African monsoon region is only 55% of the raw projection, while in South Asia and East Asian monsoon regions, the corresponding result is 71% and 76%, respectively. Hence, less land area in AfroASM will experience a significant increase of precipitation and runoff in the constrained projection. These imply that the characteristics of precipitation change in the future will be milder than the raw projection.”

L219-L222:

“The less increase of potential water availability than the raw projection may

pose a challenge to climate change adaptation and mitigation activities related to water management and food security^{48,49}, although a smaller than expected increase in rainfall will also reduce the risk of extreme precipitation and flooding.”

Figure 1. Projected changes in the Afro-Asian summer (June, July, August, and September) monsoon (AfroASM) precipitation and uncertainty of the projected changes. (a) Time series of 5-year running mean of AfroASM precipitation anomalies (units: mm day^{-1}), relative to 1950~1980 mean. Historical (grey) and SSP5-8.5 (red) simulations are shown for the 5th and 95th percentiles across 30 models (shading), and the ensemble mean (thick solid lines). The blue solid line is the AfroASM precipitation anomalies after emergent constraint. The black solid and dash lines are the observational series from the Climatic Research Unit (CRU) Time-Series (TS) version 4.02, Global Precipitation Climatology Centre version 7 (GPCC v7). (b)

Changes in precipitation (units: mm day⁻¹) under SSP5-8.5 scenario (2050-2099) relative to historical simulation (1965-2014). The region surrounded by the contour is the Afro-Asian monsoon region (see Methods). (c) The inter-model standard deviation of projected precipitation changes. Hatched regions denote signal-to-noise ratio between the absolute value of projected changes and the standard deviation less than 1.5. The regions where precipitation changes are lower than 0.1 mm day⁻¹ or over ocean is omitted.

Reference:

Cho, C., Li, R., Wang, S. Y., Yoon, J. H. & Gillies, R. R. Anthropogenic footprint of climate change in the June 2013 northern India flood. *Clim. Dyn.* **46**, 797–805 , doi: 10.1007/s00382-015-2613-2 (2016).

Kundzewicz, Z. W. *et al.* Flood risk and its reduction in China. *Adv. Water Resour.* **130**, 37–45 , doi: 10.1016/j.advwatres.2019.05.020 (2019).

Li, C., Chai, Y., Yang, L. & Li, H. Spatio-temporal distribution of flood disasters and analysis of influencing factors in Africa. *Nat. Hazards* **82**, 721–731, doi: 10.1007/s11069-016-2181-8 (2016).

Ranasinghe, R. *et al.* Chapter 12: Climate change information for regional impact and for risk assessment. in *Climate Change 2021: The Physical Science Basis. Contribution of Working Group I to the Sixth Assessment Report of the Intergovernmental Panel on Climate Change* (eds. Masson-Delmotte, V. *et al.*) (Cambridge University Press, 2021).

Singh, O. & Kumar, M. Flood events, fatalities and damages in India from 1978 to 2006. *Natural Hazards* **69**, 1815–1834, doi: 10.1007/s11069-013-0781-0 (2013).

Minor comments:

1. Use of percentages: I found phrasing such as “26% less increase” (line 77) confusing. Does this mean the precipitation increase of the constrained projection is 74% of that of the raw projections? It would help to be explicit how you are comparing the constrained and raw precipitation somewhere in the main text.

Thanks. Yes. To avoid misleading, we have rephrased relevant statement of percentages throughout the revised manuscript. Please see L25-L27 and L164 in the revised manuscript.

2. Line 65: You discuss coherent rainfall variability “on various timescales” across North Africa and Asia. Although this is true on millennial timescales, on shorter timescales the picture is less clear, and your phrasing does not specify what timescales you refer to. I suggest adjusting this line to separately cite the work looking at

millennial and decadal timescales.

Thanks. We have separately cited the work looking at different timescales. Please see L63-L66 in the revised manuscript. We also list the revised statement below:

L63-L66: “Given the fact that the monsoon precipitation in West Africa and Asia shows in-phase changes due to the modulation of interhemispheric thermal contrast (ITC) and sea surface temperature (SST) variation of North Atlantic on millennial^{1,27}, centurial^{4,12,26}, and decadal timescales^{3,28}, ...”

3. Line 84: SSP5-8.5 - acronym should probably be expanded? And perhaps it’s worth stating this is a severe scenario for anyone not familiar with the CMIP6 experimental design.

Thanks. We have expanded the SSP5-8.5 acronym, and added a brief introduction of SSP5-8.5 in the revised manuscript. Please see L77-L81 in the revised manuscript.

4. Line 88-89: Clarify that “ensemble mean and inter-model standard deviation” refer to the projected change.

Thanks. Clarified. Please see L83-L85 in the revised manuscript.

5. Line 149: You could discuss here which models/model families particularly overestimate ITC1.

Thanks. We have shown the ITC_I of each model and discussed the models which overestimate the ITC_I . Please see L154-L156 and Supplementary Table S3 in the revised manuscript. We also list this added information below.

L154-L156: “The models with a larger ECS, such as CanESM5 and CanESM5-CanOE, simulate a stronger ITC_I (Fig. 3a and Supplementary Tab. S3).”

Supplementary Table S3. The interhemispheric thermal contrast index (ITC_I) in the historical simulation of 30 CMIP6 models and in the observation for the period of 1981-2014. The red boxes denote that the simulated ITC_I exceed the range of the observation. The observed ITC_I is shown by the mean across multiple observation datasets and the uncertainty range contributed from multiple observation datasets and internal variability.

Models and Observation	ITC_I ((34K yr ⁻¹) ²)
Observation	0.12±0.03
ACCESS-CM2	0.18
ACCESS-ESM1-5	0.23

AWI-CM-1-1-MR	0.19
BCC-CSM2-MR	0.16
CAMS-CSM1-0	0.10
CESM2-WACCM	0.25
CNRM-CM6-1	0.13
CNRM-ESM2-1	0.13
CanESM5	0.30
CanESM5-CanOE	0.33
EC-Earth3	0.29
EC-Earth3-Veg	0.18
FGOALS-f3-L	0.14
FGOALS-g3	0.19
GFDL-CM4	0.22
GFDL-ESM4	0.14
GISS-E2-1-G	0.10
HadGEM3-GC31-LL	0.21
INM-CM4-8	0.12
INM-CM5-0	0.14
IPSL-CM6A-LR	0.20
MCM-UA-1-0	0.13
MIROC6	0.14
MIROC-ES2L	0.12
MPI-ESM1-2-HR	0.10
MPI-ESM1-2-LR	0.12
MRI-ESM2-0	0.14
NESM3	0.24

NorESM2-LM	0.21
UKESM1-0-LL	0.25

6. Line 164-165: “a remarkable drying tendency is seen in the observations in the latter half of the 20th century” - perhaps specifically guide the reader to compare rainfall in the 1950s with 1980s. Because of where the 0mm/day line sits, at first glance the weakening of precipitation in the 1980s did not look large to me.

Thanks. We have replaced the baseline period of 1965-2014 with the period of 1950-1980 in the Figure 1a in the revised manuscript, to highlight the drying tendency in the observation in the latter half of the 20th century. Please see the Figure 1a in the revised manuscript.

7. Typos/grammar

Line 18: Afro-Asian summer monsoon -> The Afro-Asian summer monsoon

Thanks. Corrected.

Line 18: populations -> people

Thanks. Corrected.

Line 25: larger trend -> a larger trend

Thanks. Corrected.

Line 25: more increase -> a greater increase

Thanks. Corrected.

Line 26: present-day ITC trend -> the present-day ITC trend

Thanks. Corrected.

Line 27: emergent constraint -> this emergent constraint

Thanks. Corrected.

Line 29: significant increase -> a significant increase

Thanks. Corrected.

Line 33: global monsoon system -> the global monsoon system

Thanks. Corrected.

Line 34: Sahel -> the Sahel

Thanks. Corrected.

Line 34: Indian -> the Indian

Thanks. Corrected.

Line 37: populations -> people

Thanks. Corrected.

Line 38: large spread -> a large spread

Thanks. Corrected.

Line 54: present day -> the present day

Thanks. Corrected.

Line 54: future -> the future

Thanks. Corrected.

Line 79: fractions -> fraction

Thanks. Corrected.

Line 85: western African monsoon -> the western African monsoon

Thanks. Corrected.

Line 86: increases -> increase

Thanks. Corrected.

Line 89: 90% Afro-Asian monsoon -> 90% of the Afro-Asian monsoon

Thanks. Corrected.

Line 96: systematically -> systematic

Thanks. Corrected.

Line 103: seasonal swing -> the seasonal swing

Thanks. Corrected.

Line 125: Somali -> Somalia, or refer to enhancement of the Somali jet

Thanks. Corrected to Somalia.

Line 127: Hence a larger increase of ITC in the projection... -> Hence a larger projected increase of ITC...

Thanks. Corrected.

Line 164: remarkably -> remarkable

Thanks. Corrected.

Line 305: "The SNR is multiplied by" - should this be "the regression coefficient is multiplied by"?

Yes. Corrected.

Line 307: constraint -> constrained

Thanks. Corrected.

Reviewer #3 (Remarks to the Author):

Summary

The authors analyze 21st century changes in precipitation over land across the African and Asian monsoon sectors in CMIP6 models under a high-emissions scenario and attempt to develop an emergent constraint thereof based on the interhemispheric thermal contrast. They argue that this constraint narrows uncertainty and results in a mean projection of increased precipitation over these regions, but less of an increase than without the emergent constraint.

There are several interesting results here, but the manuscript has two critical problems that would need to be fixed before being a plausible candidate for publication: (1) its lack of disentangling the effect of ECS on the precip changes of interest, and (2) confusing and potentially problematic methods employed to generate the emergent constraint. These and more minor comments are detailed below.

We would like to express our appreciation to the reviewer for the constructive comments that helped us to improve the manuscript. According to the reviewer's suggestions, we have thoroughly revised the manuscript. We believe that the new results from constraining the hydrological sensitivity and global mean warming separately and clarifying the procedures on the emergent constraint help to further convince readers of our conclusions.

Major comments

1. Role of ECS

The manuscript notes that the projected summer rainfall changes of interest are well correlated with model equilibrium climate sensitivity (ECS). This is not entirely surprising, as it is well established that more generally hydrological cycle changes to first order scale with the global-mean warming. Held and Soden 2006 and Allen and Ingram 2002 are early and canonical references here, but there is a whole literature on "pattern scaling" that discusses this in more detail, e.g. [tebaldi_pattern_2014].

Given that, as is standard in analyses of precip change under mean temperature change (I can't think of an example reference here, but there are plenty), the precip changes need to be normalized by the global-mean warming. This removes the direct influence of ECS and results in a clean disentangling of two presumably independent effects: (1) how much warming there is, and (2) given the amount of warming, how the monsoon precip will change. One can then address each factor separately, with constraints on ECS (e.g. very exhaustively addressed by [sherwood_assessment_2020]) and separately constraints such as what the present study attempts on the precip change per unit warming.

In other words, without normalizing by mean warming in each model, the present

manuscript's attempt nominally to constrain Afro-Asian monsoon precip change implicitly ends up being a muddled attempt to constrain both ECS and hydrological sensitivity over the Afro-Asian monsoon region.

Thanks for your valuable comments. As you suggested, we have normalized the precipitation changes using the global-mean warming, viz. hydrological sensitivity. The dominant pattern of projected uncertainty in hydrological sensitivity is shown below as Figure R2, which is similar to that in Figure 2, with a spatially consistent increase in precipitation over AfroASM (Figure R2a) and a “NH warmer than SH” pattern (Figure R2b and R2c). The intermodel spread of hydrological sensitivity are significantly correlated with the projected increase of ITC (Figure R3a). Based on the relationship between present-day ITC_I and the projected hydrological sensitivity, we constrain the projected hydrological sensitivity (Figure R4). The result shows that the constrained response of hydrological sensitivity is only 87% of that of the raw projection.

While global mean surface air temperature (GMSAT) warming in the constrained projection will be weaker than that in the raw projection (Tokarska et al., 2020; Liang et al., 2020; Ribes et al., 2021; Lee et al., 2021), our results show that the observational constraint leads to a weaker response of hydrological sensitivity.

We quantify the constrained projection of precipitation increase based on the constrained hydrological sensitivity and constrained GMSAT (Fig. R5). The spatial pattern of constrained projection shown in Figure R5 is like that of constraining precipitation changes directly in Figure 5 in the revised manuscript.

To quantify the relative contribution of constrained hydrological sensitivity and GMSAT to the total constrained effect, we decompose the precipitation changes as follow:

$$\Delta Pr_{Con} = \alpha_{Con} \cdot \Delta GMSAT_{Con}, \quad (1)$$

where ΔPr is the projected fractional change of precipitation (%), α is the hydrological sensitivity ($\% K^{-1}$), and $\Delta GMSAT$ is the projected GMSAT changes (K). Δ denote the projected changes in 2050-2099 relative to 1965-2014. X_{Con} and X_{Raw} denote constrained and raw projection, respectively. Since the constrained projection consists of raw projection (X_{Raw}) and constrained effect (X'), the Eq. (1) can be further divided as follow:

$$\Delta Pr_{Con} = \alpha_{Raw} \cdot \Delta GMSAT_{Raw} + \alpha_{Raw} \cdot \Delta GMSAT' + \alpha' \cdot \Delta GMSAT_{Raw} + \alpha' \cdot \Delta GMSAT', \quad (2)$$

where the $\alpha_{Raw} \cdot \Delta GMSAT_{Raw}$ denotes precipitation increase in the raw projection (ΔPr_{Raw}). The difference between constrained and raw projection is as follow:

$$\frac{\Delta Pr_{Con} - \Delta Pr_{Raw}}{\Delta Pr_{Raw}} = \frac{\alpha'}{\alpha_{Raw}} + \frac{\Delta GMSAT'}{\Delta GMSAT_{Raw}} + \frac{\alpha' \cdot \Delta GMSAT'}{\Delta Pr_{Raw}}, \quad (3)$$

The difference of precipitation increases and hydrological sensitivity between

constrained and raw projection (viz, $\frac{\Delta Pr_{Con} - \Delta Pr_{Raw}}{\Delta Pr_{Raw}}$ and $\frac{\alpha'}{\alpha_{Raw}}$) is -30% and -13%.

Following Tokarska et al. (2020) and Lee et al. (2021), we constrained the projected GMSAT warming using the observed GMSAT trend in 1981-2014. The constrained GMSAT warming is 2.62K (2.05K~3.19K, for the range of ± 1 standard deviation), weaker than the raw projection [3.30K (2.52K~4.08K)] under SSP5-8.5. The

constrained precipitation due to constrained GMSAT warming (viz. $\frac{\Delta GMSAT'}{\Delta GMSAT_{Raw}}$) is -21% of the ΔPr_{Raw} . So the constrained effects contributed from hydrological sensitivity and GMSAT reduce the raw projection by 13% and 21%, respectively.

Based on above analyses, the conclusion based on constraining hydrological sensitivity and GMSAT separately is consistent with that based on constraining the precipitation changes directly.

In addition, both the GMSAT warming and the hydrological sensitivity are closely related to the increase of ITC (Figure R3). Given that the large-scale monsoon circulation is generally driven by the ITC caused by the seasonal swing of solar incidence via diabatic fluxes of moist static energy (Trenberth et al., 2000; Geen et al., 2020; Hill, 2019), we constrain the projection of AfroASM precipitation by correcting the model biases of ITC, since a pronounced overestimation of ITC is seen in the raw projection.

In summary, since the results based on your suggestion are like our original results, instead of re-organizing the logic of the manuscript, we have added a discussion on the constraint of hydrological sensitivity and GMSAT in L223-L235 of the revised manuscript while keeping the method unchanged in the text. If you think the new analysis shown here is more convincing, we are happy to replace the original results with the new results shown above. Your recommendation is appreciated in advance. In addition, the recommended references have been cited.

We also list the discussion on the constraint of hydrological sensitivity and GMSAT below for your reference:

L223-L235: “Given that the inter-model uncertainty of global mean warming is closely associated with the inter-model spread of ECS^{50,51}, with normalizing by the global mean warming, the precipitation response (viz, hydrological sensitivity) still shows a remarkable spread across models^{10,12,52}. Recent study indicated that the projected uncertainty of hydrological sensitivity over the AfroASM regions is related to the projected uncertainty of ITC and land-sea thermal contrast¹². Since the inter-model uncertainties of both hydrological sensitivity and global mean surface air temperature (GMSAT) are significantly correlated with the ITC (Supplementary Fig. S9), we further constrain the hydrological sensitivity and GMSAT separately based on the relationship between projected uncertainty and present-day biases (see Methods). The results based on constraining hydrological sensitivity and GMSAT separately are consistent with that based on constraining the precipitation changes directly. The constrained effects contributed from

hydrological sensitivity and GMSAT reduce the raw projection by 13% and 21%, respectively.”

L409-L419: “**Constrained projection of hydrological sensitivity and GMSAT**

The hydrological sensitivity is defined as the precipitation response normalized by the GMSAT warming in 2050-2099 relative to 1965-2014. Since the inter-model spread of hydrological sensitivity is closely related to that of ITC (Supplementary Fig. S9a), which is consistent with recent study¹², we constrain the projection of hydrological sensitivity over AASM region based on Eq. (5) and Eq. (8). The constrained response of hydrological sensitivity is only 87% of that of the raw projection.

We constrained the projected GMSAT warming using the observed GMSAT trend in 1981-2014, following Tokarska⁷⁰ and Lee⁴⁰. The constrained GMSAT warming is 2.62K (2.05K~3.19K, for the range of ± 1 standard deviation), weaker than the raw projection [3.30K (2.52K~4.08K)] under SSP5-8.5.”

Figure R2. Dominant pattern of projected uncertainty of hydrological sensitivity and related historical pattern. The hydrological sensitivity is defined as the precipitation changes normalized by the global-mean warming. (a) The projected response of hydrological sensitivity (shading, $\text{mm day}^{-1} \text{K}^{-1}$) and wind at 850 hPa (UV850; vector, $\text{m s}^{-1} \text{K}^{-1}$) across 30 CMIP6 models under SSP5-8.5 scenario regress onto the inter-model normalized leading principal components (PC1). The PC1 are derived from the inter-model empirical orthogonal function (EOF) analysis of projected hydrological sensitivity under SSP5-8.5 in 2050-2099 relative to 1965-2014. The percentage on the top-right corner is explained inter-model variance. (b) the future increase of surface temperature in 2050~2099 and (c) the trend of surface temperature (K) in 1965-2014 across models regresses onto the inter-model normalized PC1. The right panels in (b) and (c) are the zonal mean of the regression

coefficient, and the thin dash vertical lines are the global area mean of the regression coefficient. The stippling, black vectors and hatching represent the regression exceeds 90% confidence level under student t-test. Black dash boxes in (c) are used to define the pattern indices to constrain the PC1.

Figure R3. Inter-model relationship between the projection changes of interhemispheric thermal contrast (ITC, K) and the hydrological sensitivity ($\% K^{-1}$), and between the projection changes of ITC and global mean surface air temperature (GMSAT, K) warming. The change and the warming are represented by the anomaly in 2050-2099 relative to 1965-2014. The results on the top-right corner are the correlation coefficient and significant level under Student t-test.

Figure R4. The scatter diagram between interhemispheric thermal contrast pattern index (ITC_1 , $(K\ 34yr^{-1})^2$) across models in the present-day climate and inter-model spread of normalized PC1 of the hydrological sensitivity. ITC_1 can explain the PC1 with high corrected correlation coefficient (r) which is shown on the top right corner. Black fitting line is obtained by the least square method, and the red fitting line is an observational correction. Dashed curves denote the 95% confidence range of the linear regression. The black vertical and horizontal dash lines denote the mean of ITC_1 across multiple observation datasets and the constrained projection, respectively. The dark gray shading denotes the range of ± 1 -time standard deviation across observation datasets. The light gray shading denotes the range contributed from the unforced internal variability.

Figure R5. The constrained projection of Afro-Asian summer monsoon

precipitation based on constrained hydrological sensitivity and global mean surface air temperature. (a) The constrained precipitation (shading, mm day⁻¹) and wind at 850 hPa (UV850; vector, m s⁻¹; vectors drawn for larger than 0.1 m s⁻¹), and (b) the constrained effect represented by the difference between constrained and unconstrained multi-model ensemble (MME).

Reference:

Geen, R., Bordoni, S., Battisti, D. S. & Hui, K. L. Monsoons, ITCZs and the concept of the global monsoon. *Reviews of Geophysics* 58, 1–60, doi: 10.1002/essoar.10502409.2 (2020).

Hill, S. A. Theories for Past and Future Monsoon Rainfall Changes. *Current Climate Change Reports* 5, 160–171, doi: 10.1007/s40641-019-00137-8 (2019).

Lee, J. Y. et al. Chapter 4: Future global climate: scenario-based projections and near-term information. in *Climate Change 2021: The Physical Science Basis. Contribution of Working Group I to the Sixth Assessment Report of the Intergovernmental Panel on Climate Change* (eds. Masson-Delmotte, V. et al.) (Cambridge University Press, 2021).

Liang, Y., Gillett, N. P. & Monahan, A. H. Climate model projections of 21st century global warming constrained using the observed warming trend. *Geophysical Research Letters* 47, doi: 10.1029/2019gl086757 (2020).

Riahi, K. et al. The Shared Socioeconomic Pathways and their energy, land use, and greenhouse gas emissions implications: An overview. *Global Environmental Change* 42, 153–168, doi: 10.1016/j.gloenvcha.2016.05.009 (2017).

Tokarska, K. B. et al. Past warming trend constrains future warming in CMIP6 models. *Science Advances* 6, doi: 10.1126/sciadv.aaz9549 (2020).

Trenberth, K. E., Stepaniak, D. P. & Caron, J. M. The global monsoon as seen through the divergent atmospheric circulation. *Journal of Climate* 13, 3969–3993, doi: 10.1175/1520-0442(2000)013<3969:tgmast>2.0.co;2 (2000).

2. Complicated or confusing procedures on the emergent constraint

There are several steps of slicing and dicing of the data before ultimately arriving at the "ITC pattern index" that is used as the emergent constraint. Perhaps these procedures are all warranted, but as written I found them confusing and as such not entirely convincing as I'll now describe.

Most importantly, the "headline" figure is Figure 4, from which the quoted values of constrained projections in the abstract are derived. But what's being "constrained" isn't actually the projected precip change in the Afro-Asian monsoon sector---it's the PC1. The behavior of the PC itself isn't ultimately what we care about---what we care

about is the actual change in rainfall over the region. But that's never shown if I'm understanding correctly; it's inferred from the constraint on the PC and its relationship to the PC.

Separately, I'm not sure that the uncertainties are being properly propagated forward into the constraint. Fig. 4 nicely includes measures of uncertainty on both the ITC index (grey shading) and in the regression line (dashed curves). In L149 the quoted "constrained" PC1 value is -0.49 ± 0.63 . But these error bounds surely underestimate the uncertainty (also it should be stated more clearly in the main text what specifically the ± 0.63 bounds are and how they were computed). I know this isn't a rigorous way of doing it, but just by eye examine in Fig. 4 the area contained within the two dashed curves and the grey shading as a plausible estimate of the uncertainty in both terms. Within that area, the intersection of the "obs" (vertical dashed line) with the projection (horizontal dashed line) could yield PC1 as low as roughly -1.4 or as high as 0.2, a considerably wider range than -0.49 ± 0.63 . If I am mistaken and the uncertainty in both the observed ITC index and the model PC1-ITC1 regression are in fact both being accounted for, I do apologize, but ask that you clarify the description in the text. If I am not mistaken, then how much does this weaken your constraint?

In addition, L137-139 describe the ITC index used as the constraint as "produced by projecting the present-day trend pattern of surface temperature onto the warming pattern associated with the inter-model PC1 shown in Figure 2c". Why do this projection step and not simply use the present-day trend pattern itself? Your procedure may indeed be the superior one, but it needs to be justified in the manuscript.

Thanks. The reason for using PC1 in Figure 4 is that we want to constrain the spatial pattern of AfroASM precipitation projection instead of a regional average. In the PC1, different model has different values, rightly representing the inter-model uncertainty. The corresponding EOF1 pattern, showing the spatial structure of uncertainty, is model independent. So constraining the PC1 is equivalent to constraining the precipitation itself. Constrained precipitation can be reconstructed by the constrained PC1 and the EOF1 (see Methods). Based on your suggestion, we have shown the constrained AfroASM precipitation directly in Figure 4a of the revised manuscript. Since the regional patterns are informative for policymakers, we not only constrain the regional mean of AfroASM precipitation but also its spatial patterns (Fig. 5a). This explains why we constrain the PC1 in Figure 4b in the revised manuscript. Please see L143-L147, L156-L158 and Figure 4 in the revised manuscript.

The spread of constrained PC1 is estimated by the intermodel variance of constrained PC1, which is compared with the variance of unconstrained PC1. We correct the PC1 of individual model using the observation of ITC_I , based on the relationship of emergent constraint. Since the PC1 have been standardized before emergent constraint, the standard deviations of constrained and unconstrained PC1 are 0.80 and 1, respectively. Hence, the variance of constrained PC1 is about 37% less than that of unconstrained PC1. On the other hand, the dark and light gray shading denotes the

range contributed from the spread across multiple observation datasets and the unforced internal variability. Similarly, the inter-model standard deviations of constrained and unconstrained AfroASM precipitation are 6.13% and 7.70%, respectively. So the constrained variance is also 37% less than the unconstrained one. As you suggested, we have described the method with more details in the revised manuscript. Please see L366-L371 in the revised manuscript.

To clearly reflect the present-day warming pattern and reduce the effect of random noise, we project the historical warming trend of surface temperature onto the PC1-related warming trend using the scalar product in Northern Hemisphere (NH; 20°N~50°N, 0~360°) and Southern Hemisphere (SH; 20°S~50°S, 0~360°), respectively, rather than use the present-day trend pattern directly, following Chen et al. (2020). In addition, we have used the present-day trend of surface temperature directly to calculate the ITC between NH- and SH- longitudinal range of the Eurasian continent (20°W~150°E), and also obtain a significant relationship between the PC1 and the trend of ITC ($r = 0.45$, $p < 0.02$).

We have justified the procedure of calculating the ITC_t . Please see L309-L316 in the revised manuscript.

Figure 4. The scatter diagram between interhemispheric thermal contrast pattern index (ITC_1 , $(K 34yr^{-1})^2$) across models in the present-day climate and inter-model spread of AfroASM precipitation (A) and normalized PC1 (B). ITC_1 can explain the PC1 with high corrected correlation coefficient (r) which is shown on the top right corner. Black fitting line is obtained by the least square method, and the red fitting line is an observational correction based on Equation (5) (Eq. (5); see Methods). Dashed curves denote the 95% confidence range of the linear regression. The black vertical and horizontal dash lines denote the mean of ITC_1 across multiple observation datasets and the constrained projection, respectively. The dark gray shading denotes the range of ± 1 -time standard deviation across observation datasets. The light gray shading denotes the range contributed from the unforced internal variability (see Methods)

Reference:

Chen, X. et al. Emergent constraints on future projections of the western North Pacific Subtropical High. Nature Communication 11, doi: 10.1038/s41467-020-16631-9 (2020).

Minor comments

1. L2: "less rich"

Thanks. We have revised the title as "Observationally constrained projection of Afro-Asian monsoon precipitation".

2. L33-34: An ITCZ-like narrow band is a reasonable description over Africa but certainly not for the Indian and E. Asian monsoons, which are more complicated

Thanks. Rephrased. Please see L33-L34 in the revised manuscript.

3. L38-39: "raising risk on failure in addressing regional climate change" confusing wording; rephrase

Thanks. Rephrased. Please see L37-L38 in the revised manuscript:

4. L53: To me, the definitive citation on emergent constraints and how to do them in the most physically justifiable way is [klein_emergent_2015]. I recommend you cite it and ensure your approach checks all of their requirements for a sound constraint.

We thank the reviewer for recommending the relevant paper. We have read and cited it in the revised manuscript. We have also checked the requirements of emergent constraint. Please see L51-L54, L116-L135, and L204-L208 in the revised manuscript.

5. L69-70: I don't think the reference to the Scenario Model Intercomparison Project is necessary; just referring to CMIP6 is sufficient and has less danger of being confusing

Thanks. Deleted. Please see L68-L69 in the revised manuscript.

6. L84: The manuscript should briefly note that SSP8.5 is now a highly unlikely scenario given recent trends in decarbonization; e.g. [hausfather_emissions_2020].

Thanks. We have added a description on SSP5-8.5 scenario in the revised manuscript, and read and cited the recommended reference very carefully. Please see L80-L81 and L252-L253 in the revised manuscript:

7. L96: systematically -> systematic

Thanks. Corrected.

8. L102-104: The manuscript should briefly note that, even if the thermal contrast proves to be a useful predictor, modern understanding of the monsoon replaces thermal gradients with moist entropy or moist static energy gradients. The Geen et al review paper cited notes this; see also e.g. [hill_theories_2019].

We thank the reviewer for recommending the relevant paper. We have read and cited it carefully in the revised manuscript, and have rephrased the statement. Please see L101-L103 in the revised manuscript.

9. L110-112: "rooted in" is too vague; please clarify. Also, from Fig. 2c there is much more noise for the historical, such that your claim is not immediately valid by eye from that plot. I recommend adding panels of zonal averages for both the future and historical to make this clearer.

Thanks. We have rephrased the "is rooted in" with "resulted from". In addition, we have added the panels of zonal averages for Figure 2b and Figure 2c. Please see L109-L110 and Figure 2 in the revised manuscript.

10. L115-116: Should cite one or more papers of the voluminous literature on transient warming, Arctic amplification, etc.

Thanks. Cited. Please see L117-L119 in the revised manuscript.

11. L123-125: These are far removed from the E. Asian monsoon sector which you otherwise are including. In addition, similar to the suggestion above for Fig. 2 I recommend adding panels of zonal-mean fields: this is quite noisy to the point that I don't find the claim about clear increases in cross-equatorial flow totally convincing.

Thanks. Both of a larger increase of ITC and a larger PC1 will induce a stronger enhancement of low-level cross-equatorial flow over South China Sea (around 105°E), which is closely linked to East Asian Summer monsoon (Supplementary Fig. S3). We have added a statement in the revised manuscript and added panels of zonal-mean fields in the Supplementary Figure S3. Please see L125-L127 and Supplementary Figure S3 in the revised manuscript.

Supplementary Figure S3. The low-level cross equator flow related to PC1 and projected interhemispheric thermal contrast (ITC). (a) The anomalies of the meridional wind (shading) and the horizontal circulation (vector) in the low-level troposphere (925 hPa, units: m s^{-1}) and (b) the longitude-height cross section of the meridional wind ($4^{\circ}\text{S}\sim 4^{\circ}\text{N}$, units: m s^{-1}) in 2050-2099 relative to 1965-2014 across models regress onto the normalized inter-model PC1. (c) and (d) is same as (a) and (b) but for the regression onto the normalized ITC changes across models in 2050-2099. The stippling and vectors denote the regression exceeds 90% confidence level. The right panels in (a) and (c) are the zonal mean of the low-level meridional wind related to PC1 and ITC, respectively.

12. Fig. 4: Omit the "(a)" label since there's only one panel.

Thanks. Omitted.

13. L250-251: I'm not familiar with these papers, but is this saying that the connection between the Afro-Asian monsoon precip change and the NH-SH thermal contrast has already been well established? If so, that needs to be stated in the main text, not just the methods, and it needs to be clarified specifically what in this study is novel (presumably all the emergent constraint material).

Thanks. These papers have mentioned the connection between the increase of monsoon precipitation changes and the increase of NH-SH thermal contrast in the projection, while they did not propose an observational constraint (Endo et al., 2018;

Cao et al., 2020; Wang et al., 2020). In this study, we identify a physically grounded constraint on the intermodal spread in projected precipitation using the observational datasets, which is contribution and innovation of this study, as the summary of Reviewer #2.

We have cited these papers in the main text very carefully, and clarified the innovation in our study. Please see L60-L68 in the revised manuscript.

References:

Cao J, Wang B, Wang B, Zhao H, Wang C, Han Y. Sources of the inter- model spread in projected global monsoon hydrological sensitivity. *Geophysical Research Letters* 47, doi:10.1029/2020gl089560 (2020)

Wang B, Jin C, Liu J. Understanding future change of global monsoon projected by CMIP6 models. *Journal of Climate* 33, 6471-6489, doi:https://doi.org/10.1175/jcli-d-19-0993.1 (2020)

Endo H, Kitoh A, Ueda H. A Unique Feature of the Asian Summer Monsoon Response to Global Warming: The Role of Different Land–Sea Thermal Contrast Change between the Lower and Upper Troposphere. *Sola* 14, 57-63, doi:10.2151/sola.2018-010 (2018)

14. L256-259: Why these latitude ranges? Why do they differ between the hemispheres? Without more physical justification, this feels a little fishy.

Thanks. To maximum the constrained effect, we define the interhemispheric thermal contrast (ITC) as the difference between (20°N~50°N, 0~360°E) and (0~20°S, 0~360°E).

Following previous studies (Cao et al., 2020; Wang et al., 2020), we define the ITC as the difference between (0~60°N, 0~360°E) and (0~40°S, 0~360°E), and have similar relationship of the emergent constraint ($r = 0.54$, $p < 0.01$; Fig. R3).

In addition, we define the ITC as the symmetrical difference between Northern Hemisphere (20°N~50°N, 0~360°) and Southern Hemisphere (20°S~50°S, 0~360°), and have similar relationship of the emergent constraint ($r = 0.61$, $p < 0.01$).

Hence, the relationship of the emergent constraint is independent from the selection of latitude range in the definition of ITC. As you suggested, we have replaced the definition of ITC with the symmetrical difference between Northern Hemisphere (20°N~50°N, 0~360°) and Southern Hemisphere (20°S~50°S, 0~360°). Please see L302-L305, L309-L314 and Figure 4 in the revised manuscript.

Figure R3. Same as Figure 3 but the ITC_1 is defined as the difference between (0~60°N, 0~360°) and (0~40°S, 0~360°).

15. L262: Can you really claim that internal variability influence is weak over a ~3 decades timescale on regional scales? C.f. Clara Deser and Karen McKinnon's work (among others) with large ensembles and internal variability.

Thanks. The contribution from the internal variability accounts for about 20% based on the variance ratio between the internal variability and intermodel spread. Please see L316-L326 in the revised manuscript.

To avoid misleading, we have rephrased our expression. Please see L306-L308 in the revised manuscript.

Bibliography

=====

- 1) [tebaldi_pattern_2014] Tebaldi \& Arblaster, Pattern Scaling: $\{\{Its\}\}$ Strengths and Limitations, and an Update on the Latest Model Simulations, *{Climatic Change}*, **122(3)**, 459--471 (2014). link. doi.
- 2) [sherwood_assessment_2020] Sherwood, Webb, Annan, Armour, Forster, Hargreaves, Hegerl, Klein, Marvel, Rohling, Watanabe, Andrews, Braconnot, Bretherton, Foster, Hausfather, von der Heydt, Knutti, Mauritsen, Norris, Proistosescu, Rugenstein, Schmidt, Tokarska \& Zelinka, An Assessment of

{{Earth}}'s Climate Sensitivity Using Multiple Lines of Evidence, *{Reviews of Geophysics}*, **n/a(n/a)**, e2019RG000678 (2020). link. doi.

- 3) [klein_emergent_2015] Klein \& Hall, Emergent {{Constraints}} for {{Cloud Feedbacks}}, *{Curr Clim Change Rep}*, **1(4)**, 276--287 (2015). link. doi.
- 4) [hausfather_emissions_2020] Hausfather \& Peters, Emissions \textendash{ } the 'Business as Usual' Story Is Misleading, *{Nature}*, **577(7792)**, 618--620 (2020). link. doi.
- 5) [hill_theories_2019] Hill, Theories for Past and Future Monsoon Rainfall Changes, *{Curr Clim Change Rep}*, **5(3)**, 160--171 (2019). link. doi.

REVIEWERS' COMMENTS

Reviewer #1 (Remarks to the Author):

Review #2: Observationally constrained projection of Afro-Asian monsoon precipitation

The authors replied to the concerns I added. I think the manuscript is better in the present form. I found the replies to others reviewers well described as well.

Nevertheless, I just would like to see a conclusion with the amount of precipitation that changes in the current mm/day unit (i.e. the prior and posterior estimates). The whole talk about % of changes, but it is nice to come back to what precipitation means.

This point is related to what emergent constraints are useful for: estimating changes in the mean, and changes in the spread. While the latter is well described, the description of the former can be unclear. A clarification would be valuable for the paper.

Nonetheless, I think this paper is worth publishing.

Reviewer #2 (Remarks to the Author):

I am happy that the reviewers have addressed my points and am happy to recommend the revised manuscript for publication with only a few minor comments:

Line 26 - 'Shows' seem strong to me when referring to an emergent constraint, I suggest 'indicates' or 'suggests'

Line 94-97 - Thanks for this additional analysis. I suggest adding '(not shown)' on line 97 for clarity that this check isn't in Fig 2.

Line 110 - 'results from' I think it would be clearer and more accurate to say 'correlates with the trend over the historical period'?

Line 111 - replace 'more' with 'a greater'

Line 157 - if 'time' is mathematical here a times sign 'x' would be clearer to avoid association with clock time

Line 180 & 219 - 'less' -> 'smaller'

Line 206 - the ensembles referred to here are subsets of the CMIP6 MME (plus the randomized outliers). I suggest making this clear e.g. 'by using different subsets of models and including randomized outliers'

Methods - Model Simulations section - number of models should come before CMIP6: 'CMIP6 30 models' -> '30 CMIP6 models', etc.

Line 289 - Add a short sentence to explain this was a sensitivity test e.g. 'To confirm results are not dependent on the strength of individual models' hydrological cycles,...'

Fig. 4a - Light gray dashed line does not seem to be described in the caption

Reviewer #3 (Remarks to the Author):

Second round review of ``Observationally constrained projection of afro-asian monsoon precipitation''

Manuscript authors: Ziming Chen et al

Summary

I thank the authors for their thorough revision efforts, which have definitely improved the manuscript. However, I interpret them as largely confirming my suspicions regarding the conflation of mean warming with the hydrological sensitivity. In particular, compare Fig. 4b in the new manuscript to Fig. R4 in the reply to the reviewers: the correlation coefficient is reduced by nearly 40% when normalizing by the mean warming! So, even if the p-value technically remains "significant" based on e.g. the conventional 0.05 threshold, this suggests to me that your focus on the non-normalized response in the main text, with the normalized response relegated to the supplemental material, is misguided.

Ultimately, I feel similarly to how another reviewer put it in the previous round: I'm skeptical of the importance of the results, but nevertheless I suppose the methods are reasonable enough that the work should ultimately be published. And my arguments are more about emphasis and framing than with the validity of the methods etc. I suspect others will find the results more compelling. So I think it is suitable for Nature Communications pending minor revisions listed below (some of which are just comments).

Line-by-line comments

L30

"will reduce" is too certain; consider "will likely reduce"

L47-59

I don't understand this sentence; how does "model uncertainty" explain a quantifiable fraction (apparently 70%) of the "inter-model variance"?

L60-64

Consider splitting into two sentences: "Recent studies have reported a connection between the uncertainty of AfroASM precipitation changes and the increase of interhemispheric thermal contrast^{4,12,13,27}\$. But how to constrain the projection and reduce the spread remains unknown."

L68

omit "hope to"

L72-73

Please make Question 3 more precise. What do you mean by "impacts" (a word that has multiple meanings)?

L101-103

I still think this framing, in which the thermal contrast is the most fundamental driver, is incorrect. Even if you note that it is influenced by MSE fluxes, it is the MSE gradients themselves that are more fundamental. That being said unless relative humidity has strong spatial structure distinct from temperature, the dry and moist tracers tend to go together. See e.g. Zhou, Wenyu, and Shang-Ping Xie. "A Hierarchy of Idealized Monsoons in an Intermediate GCM." *Journal of Climate* 31 <https://doi.org/10.1175/JCLI-D-18-0084.1>.

L180,219

less increase -> smaller increase

Fig. 2 (and corresponding text elsewhere)

- Thank you for adding the zonal averages panels. Though in your response you seem to interpret them otherwise, to me these make even clearer the lack of correspondence between the historical and SSP patterns, in the Southern Ocean especially!

Fig. 3 (and corresponding text elsewhere)

- I realize now that I should have emphasized the transient climate response (TCR), rather than the

equilibrium climate sensitivity (ECS), regarding the global-mean temperature change. TCR is more directly relevant to the first-century transient response than is ECS, since the latter is influenced by centennial and millennial timescale processes. But both of these metrics are less relevant ultimately than the actual model output of the global-mean surface temperature change averaged over the time period of interest, which is what you rightly used.

- Unconstrained MME
- + ACCESS-CM2
- + ACCESS-ESM1-5
- + AWI-CM-1-1-MR
- + BCC-CSM2-MR
- + CAMS-CSM1-0
- * CanESM5
- * CanESM5-CanOE
- * CESM2-WACCM
- * CNRM-CM6-1
- * CNRM-ESM2-1
- EC-Earth3
- EC-Earth3-Veg
- FGOALS-f3-L
- FGOALS-g3
- GFDL-CM4
- GFDL-ESM4
- GISS-E2-1-G
- HadGEM3-GC31-LL
- INM-CM4-8
- INM-CM5-0
- △ IPSL-CM6A-LR
- △ MCM-UA-1-0
- △ MIROC6
- △ MIROC-ES2L
- △ MPI-ESM1-2-HR
- ◇ MPI-ESM1-2-LR
- ◇ MRI-ESM2-0
- ◇ NESM3
- ◇ NorESM2-LM